PREPARED FOR SUBMISSION TO JHEP

# Boundary RG Flows for Fermions and the Mod 2 Anomaly

**Philip Boyle Smith and David Tong**

*Department of Applied Mathematics and Theoretical Physics,*
*University of Cambridge, Cambridge, CB3 0WA, UK*

*E-mail:* pb594@damtp.cam.ac.uk, d.tong@damtp.cam.ac.uk

ABSTRACT: Boundary conditions for Majorana fermions in $d = 1+1$ dimensions fall into one of two SPT phases, associated to a mod 2 anomaly. Here we consider boundary conditions for $2N$ Majorana fermions that preserve a $U(1)^N$ symmetry. In general, the left-moving and right-moving fermions carry different charges under this symmetry, and implementation of the boundary condition requires new degrees of freedom, which manifest themselves in a boundary central charge $g$.

We follow the boundary RG flow induced by turning on relevant boundary operators. We identify the infra-red boundary state. In many cases, the boundary state flips SPT class, resulting in an emergent Majorana mode needed to cancel the anomaly. We show that the ratio of UV and IR boundary central charges is given by $g_{IR}^2/g_{UV}^2 = \dim \mathcal{O}$, the dimension of the perturbing boundary operator. Any relevant operator necessarily has $\dim \mathcal{O} < 1$, ensuring that the central charge decreases in accord with the $g$-theorem.

# 1   Introduction

Quantum field theories with boundaries are interesting for many reasons, from the role of edge modes in condensed matter physics, to impurity problems, to D-branes in string theory.

In this paper we return to an old and well explored subject: boundary conditions for free, massless fermions in $d = 1 + 1$ dimensions. As we review below, given such a collection of fermions there are an infinite number of boundary conditions that one can impose. Typically, these boundary conditions involve the introduction of new degrees of freedom at the boundary. At low energies, below any interaction scale, the number of such degrees of freedom is captured by a boundary central charge $g$, first introduced by Affleck and Ludwig [1].

The $d = 0 + 1$ dimensional boundary behaves, in many ways, like any other quantum field theory. There are operators restricted to the boundary and these can be classified as relevant, irrelevant or marginal. Operators that are exactly marginal move among a continuous family of boundary conditions. Meanwhile, boundary operators that are relevant initiate an RG flow within the space of boundary conditions without endangering the gapless nature of the bulk modes. As in higher dimensional situations, the number of boundary degrees of freedom $g$ necessarily decreases under RG flow [2, 3].

The purpose of this paper is to study such RG flows between different boundary conditions for massless fermions. We will find a simple, and elegant story in which, with some reasonable assumptions, one can follow boundary RG flows from one fixed point to another. There are a number of different aspects to this story, not least the fact that boundary conditions for fermions are classified by a $\mathbb{Z}_2$ anomaly, and so fall into one of two different classes. In this extended introduction, we review this $\mathbb{Z}_2$ anomaly before summarising our main results.

## 1.1   The Mod 2 Anomaly

A single Majorana fermion in quantum mechanics provides what is arguably the simplest system suffering an anomaly. To see this, we can start by taking two copies of a Majorana fermion, $\lambda_1$ and $\lambda_2$. Canonical quantisation gives rise to a 2d Clifford algebra $\{\lambda_i, \lambda_j\} = \delta_{ij}$ which acts irreducibly on a Hilbert space of dimension 2. This means that a single Majorana fermion would act on a Hilbert space of dimension $\sqrt{2}$, which is nonsensical.

Indeed, the dimension of the Hilbert space is counted by the path integral for a single Majorana mode, with anti-periodic boundary conditions in the temporal direction. This can be computed and is given by

$$Z_{\text{Majorana}} = \text{Tr}_{\mathcal{H}}(\mathbb{1}) = \sqrt{2} \tag{1.1}$$

This reflects the fact that there is no way to consistently quantise a single Majorana mode in $d = 0 + 1$ dimensions. This simple fact is the essence of the mod 2 anomaly, and the telltale factor of $\sqrt{2}$ will be a recurring motif throughout this paper.

As described in [4, 5], this same anomaly is lurking when we attempt to place fermions in $d = 1+1$ dimensions on a manifold with boundary. (A beautifully clear explanation of this from the continuum perspective can be found in the talk [6].) Consider a single, massive Majorana fermion $\chi$, now in $d = 1 + 1$ dimensions. There are two possible boundary conditions that one can impose, reflecting the left-moving fermion $\chi_L$ into the right-moving fermion $\chi_R$,

$$\chi_L = \pm\chi_R \tag{1.2}$$

Solving the Dirac equation, one finds that for one choice of sign there is a single Majorana zero mode localised on the boundary, while for the other there is not. The sign choice that gives rise to the zero mode is therefore inconsistent unless something else comes to the rescue to cancel the anomaly. (Which boundary condition suffers a zero mode depends on both the sign of the fermion mass, and the orientation of the boundary.)

The anomaly manifests itself in a slightly different way when we consider a complex, Dirac fermion $\psi = \chi_1 + i\chi_2$. There is no problem if we impose a boundary condition that preserves the vector $U(1)_V$ symmetry,

$$V: \quad \psi_L = \psi_R \tag{1.3}$$

This translates to the same sign (1.2) on both $\chi_1$ and $\chi_2$. This means that, if $\psi$ is massive in the bulk, then there are either two boundary zero modes or none. Either way, the system does not suffer an anomaly.

In contrast, we could impose boundary conditions of the form

$$A: \quad \psi_L = \psi_R^\dagger \tag{1.4}$$

Such boundary conditions arise in wires attached to superconductors, where an incident electron rebounds as a hole, a process known as *Andreev reflection.* If the bulk fermion

is massless, then the Andreev boundary conditions preserve the $U(1)_A$ axial symmetry of the fermion. We will consider such massless bulk fermions shortly, but for now it will be useful to keep the fermion massive. In this case, the discussion above tells us that we have a problem: the two Majorana fermions $\chi_1$ and $\chi_2$ have opposite signs for their boundary conditions, meaning that one has a zero mode and the other does not. The axial boundary condition is anomalous.

There are various ways of dealing with this. One obvious approach is simply to add by hand a quantum mechanical Majorana mode $\lambda$, which then pairs with the zero mode to render the theory consistent.

Alternatively, the anomaly can be cancelled through an inflow mechanism [7]. On a $d = 1 + 1$ Riemann surface without boundary, but endowed with a spin structure, there exists a particular SPT phase whose partition function is given by $(-1)^{\mathrm{Arf}}$, where the Arf invariant takes values $\pm 1$ depending on whether the spin structure is even or odd. This SPT phase arises, for example, as the infra-red limit of two Majorana fermions with masses of opposite sign and, in the condensed matter literature, it is better known as the topological phase of the Kitaev chain [5, 8] . Recent applications of this topological field theory can found [9–20]. However, on a Riemann surface with boundary, the Arf topological field theory is not well defined: it suffers the same mod 2 anomaly that we saw above. This anomaly can be cancelled if we have a single Dirac fermion $\psi$ living on the Riemann surface and impose the boundary condition (1.4).

The upshot of this is that, if we chose not to add further Majorana modes by hand, then a trivial bulk theory requires that we impose the vector boundary condition (1.3), while the non-trivial SPT phase requires that we impose the axial boundary condition (1.4). Note, in particular, that on a finite cylinder it is inconsistent to impose vector boundary conditions on one end, and axial boundary conditions on another.

The story above was told for massive bulk fermions. It is convenient to introduce such a mass because it makes the $\mathbb{Z}_2$ anomaly manifest in the presence of normalisable Majorana zero modes. However, anomalies are famously independent of the mass, and our $\mathbb{Z}_2$ anomaly is no different. This means that the boundary conditions (1.3) or (1.4) are also dictated by the bulk topological SPT phase for massless fermions.

**An Application to D-Branes**

A particularly elegant application of the discussion above can be found in the context of D-branes in string theory [4, 6, 21, 22]. Although not directly relevant for our story, it is lovely enough to warrant a quick advertisement.

First, various GSO projections, which characterise the different types of string theories, arise from the inclusion of various Arf invariants on the string worldsheet. When the dust settles, one finds familiar results, viewed through a new lens. The fact that BPS D-branes in Type IIA string theory have odd worldvolume dimension, while those in Type IIB have even worldvolume dimension can be traced to the Arf invariants on the worldsheet, which put different restrictions on the number of worldsheet fermions that obey the boundary conditions (1.3) and (1.4)

Furthermore, both Type IIA and Type IIB string theories are known to have non-BPS D-branes whose worldvolume dimensions are the complement of the BPS D-branes. To avoid the $\mathbb{Z}_2$ anomaly, the end point of the string must necessarily come with an extra Majorana mode. This provides a unified explanation for a number of previously observed properties of non-BPS D-branes, including the fact that their tension is a factor of $\sqrt{2}$ larger than their BPS counterparts [23, 24]. This $\sqrt{2}$ can be traced directly to the partition function (1.1) of the excess Majorana mode.

## 1.2   Chiral Boundary Conditions

Our interest in this paper lies in boundary conditions for multiple massless fermions. Here there are many more possibilities, ones that do not involve simple repetitions of the boundary conditions (1.2), (1.3) and (1.4).

These novel boundary conditions can be distinguished by the symmetries that they preserve. The two boundary conditions (1.3) and (1.4) preserve the $U(1)_V$ and $U(1)_A$ symmetry of a single, massless Dirac fermion respectively. However, in general it is possible to impose boundary conditions that preserve chiral symmetries, under which the left- and right-moving fermions carry different charges. Indeed, there is a general expectation that one can impose boundary conditions preserving any symmetry that does not suffer a 't Hooft anomaly. (See, for example, [25, 26].)

For example, if we have $N$ left-moving Weyl fermions in $d = 1 + 1$ with charges $Q_i$ under a $U(1)$ symmetry, and $N$ right-moving fermions with charges $\bar{Q}_i$, then one can impose boundary conditions that preserve the $U(1)$ symmetry provided that

$$\sum_{i=1}^{N} Q_i^2 = \sum_{i=1}^{N} \bar{Q}_i^2$$

which is the requirement that this symmetry does not suffer a 't Hooft anomaly.

There is no way to impose such boundary conditions directly on the fermion fields in the Lagrangian. Instead, one should introduce new boundary degrees of freedom, which interact with the fermions, typically in a strongly coupled fashion. However, in the far infra-red, any boundary condition for massless, bulk fermions in $d = 1 + 1$ dimensions can be encoded in a conformal boundary state [27]. The degrees of freedom necessary to impose chiral boundary conditions now show up as a contribution to the boundary central charge $g$ [1]. In this paper, we work with such conformal boundary states, a technology that we review in Section 2. The relationship between SPT phases and conformal boundary field theory was previously explored in [28–31].

To our knowledge, the general class of boundary states for $2N$ massless Majorana fermions is not known[1]. To make progress, we will restrict ourselves to boundary conditions which preserve a manifest $U(1)^N$ symmetry[2]. It is then straightforward to construct the boundary state preserving your favourite chiral, non-anomalous symmetry. Early examples of such states were introduced in [33, 34].

Given the discussion of the mod 2 anomaly in the previous section, the first question that we should ask is: into what class does a given boundary state fall? Does it describe a boundary condition that is allowed in the trivial bulk theory, or in the SPT phase? This was answered in [35] where it was shown that all chiral boundary states do indeed fall into two, mutually incompatible, classes that, following the notation of (1.3) and (1.4), we denote as *vector* and *axial*.

There is a slightly different perspective that one can take on this. As explained in [28], there is a close connection between conformal boundary states and the gapped phases of a theory. Specifically, one could consider turning on a gapping interaction only in one half of space. Low energy excitations incident from the gapless phase will then be reflected, experiencing the gapped half-space as a conformal boundary condition. Yet, as we have described above, there is a $\mathbb{Z}_2$ classification of such fermionic SPT phases: trivial and non-trivial, where non-trivial means $(-1)^{\mathrm{Arf}}$. The vector and axial classification of boundary states tells us whether these boundary states arise from trivial (vector) or non-trivial (axial) gapped phases.

---

[1]In the special case $N = 1$, the complete classification is known [37, 38]. In addition to the vector and axial states there is an interval's worth of extra states [39] that interpolate between superpositions of states in different classes, and so appear to be ruled out as pathological, at least from the perspective of SPT phases.

[2]We impose this requirement as a necessary crutch that allows us to construct the boundary states. The full symmetry group may be larger than $U(1)^N$; the conditions under which such an enhancement occurs are detailed in [32].

**RG Flows: A Summary of Our Results**

The purpose of this paper is to describe the boundary RG flows between different chiral boundary states when we perturb by a relevant operator. Any such relevant perturbation necessarily breaks one or more of the $U(1)^N$ symmetries. However, we propose that, while the RG flow breaks the symmetry, a new emergent $U(1)^N$ symmetry is restored at the end of the RG flow. It is not obvious that this is the case: one might have anticipated that, by flowing away from states preserving a full $U(1)^N$ symmetry, we would leave them for good. Instead we argue that, like the famous hotel, you can check out from these states, but you can never leave.

Assuming that a full $U(1)^N$ emerges in the infra-red allows us to track the RG flow. There are a number of interesting features that emerge from our analysis. First, one can ask: is it possible to flow from one class of boundary states to the other? Say, from vector-like boundary conditions to axial-like boundary conditions? Given the anomaly restrictions described above, one might have thought that such flows are forbidden. Instead, we find that they are very much allowed. However, whenever such a flow occurs, the resulting boundary state comes equipped with an extra Majorana mode $\lambda$, needed to cancel the anomaly.

Secondly, we find the following surprising and simple formula: if we initiate an RG flow by turning on a single, relevant boundary operator $\mathcal{O}$ with dimension $\dim \mathcal{O}$, then the UV and IR central charges are related by

$$g_{IR} = g_{UV} \sqrt{\dim \mathcal{O}} \tag{1.5}$$

Since a relevant boundary operator necessarily has $\dim \mathcal{O} < 1$, this relation is consistent with the $g$-theorem [1–3], which states that the boundary central charge $g$ must decrease.

## 1.3 The Plan of the Paper

We start in Section 2 by reviewing the construction of boundary states that preserve chiral symmetries. We also take this opportunity to introduce our notation. In Section 3 we compute the partition function for free fermions on an interval, with the same boundary state imposed on each end. This allows us to determine the spectrum of boundary operators and, in particular, extract the possible relevant boundary operators for each symmetry.

The RG analysis is given in Section 4. We explain how, for each relevant boundary operator, there is a unique candidate for the end-point of the flow, and elaborate on a

number of subtleties that arise including the emergence of Majorana bound states and what string theorists refer to as Chan-Paton factors. The statements of the results are more straightforward than the proofs; these statements are placed front and centre, and we refrain as long as possible from wallowing in the glorious technicalities. The wallowing finally occurs in Section 5.

### A Slightly Different Application to D-Branes

As far as we are aware, the kinds of chiral boundary conditions that we discuss do not have application to the fermions on the superstring worldsheet. However, there is a more indirect connection. We could consider bosonizing our fermions so that the chiral boundary conditions now describe the end-point of a string moving on a torus $\mathbf{T}^N$, with radius of order the string length.

In this context, the chiral boundary conditions are nothing more than D-branes in bosonic string theory, wrapping $\mathbf{T}^N$ with fluxes. Even translated to this familiar context, our results appear novel. Things are simplest for $N = 2$ fermions, corresponding to a D2-brane wrapping $\mathbf{T}^2$. After a T-duality, the general chiral boundary state simply translates to a D-string wrapped $(p, q)$ times around the two cycles of $\mathbf{T}^2$. We describe this in Appendix C.

## 2 Chiral Boundary States

In this section we describe the general set-up, and the symmetries that we wish to preserve in the presence of a boundary.

Our starting point is the theory of $2N$, free Majorana fermions in $d = 1 + 1$ dimensions. When this theory is placed on a spatial manifold without boundary, these fermions have a $O(2N)_L \times O(2N)_R$ global symmetry, independently rotating the left- and right-moving Majorana-Weyl fermions. However, in the presence of a boundary, this symmetry group is necessarily reduced.

A particularly straightforward class of boundary conditions can be implemented by imposing linear restrictions on the fermionic fields, such as (1.2), (1.3) or (1.4). However, these are not the most general class of boundary conditions. Instead, the generic boundary condition does not arise by restricting the value of the field on the boundary; instead it arises by imposing certain conditions on currents.

We will ask that the boundary preserves a subgroup

$$U(1)^N \subset U(1)_L^N \times U(1)_R^N \subset SO(2N)_L \times SO(2N)_R \qquad (2.1)$$

The left-moving and right-moving fermions are assigned charges $Q_{\alpha i}$ and $\bar{Q}_{\alpha i}$ respectively, where $i = 1, \ldots, N$ labels the species of complex fermion, while $\alpha = 1, \ldots, N$ labels the $U(1)$ symmetry group. The simple linear boundary conditions described above arise, for example, if $Q = \pm\bar{Q}$. Our interest in this paper lies in the more interesting boundary conditions in which the left- and right-moving fermions carry different charges. These are chiral boundary conditions.

It is not true that any choice of $U(1)^N$ symmetry can be preserved by the boundary. Only those symmetries that do not suffer a 't Hooft anomaly give suitable boundary conditions. (See, for example, [25, 26].) This means that the charge matrices necessarily obey the condition

$$Q_{\alpha i} Q_{\beta i} = \bar{Q}_{\alpha i} \bar{Q}_{\beta i} \qquad (2.2)$$

We will need a few further objects constructed from these charges. First, we introduce

$$\mathcal{R}_{ij} = (\bar{Q}^{-1})_{i\alpha} Q_{\alpha j}$$

This rational, orthogonal matrix will be sufficient to encode the charges preserved by the boundary. The boundary condition (1.3) in which each left-moving fermion is reflected into a right-mover corresponds to $\mathcal{R} = \mathbb{1}$. Imposing Andreev reflection (1.4) on each fermion corresponds to $\mathcal{R} = -\mathbb{1}$.

We also associate a charge lattice $\Lambda[\mathcal{R}] \subseteq \mathbb{Z}^N$ to our choice of boundary condition. This is defined by

$$\Lambda[\mathcal{R}] = \left\{ \lambda \in \mathbb{Z}^N \ : \ \mathcal{R}\lambda \in \mathbb{Z}^N \right\} \qquad (2.3)$$

In words: the lattice $\Lambda[\mathcal{R}]$ consists of all integer-valued vectors $\lambda \in \mathbb{Z}^N$ which remain in $\mathbb{Z}^N$ when rotated by the rational matrix $\mathcal{R}$. As we will see, this lattice plays an important role in our story.

For both standard and Andreev boundary conditions, this lattice is simply $\Lambda[\mathcal{R} = \pm\mathbb{1}] = \mathbb{Z}^N$. For chiral boundary conditions, the lattice is sparser and more interesting.

## 2.1 Constructing Boundary States

We wish to construct boundary conditions that preserve a chiral $U(1)^N$ symmetry. The key idea is due to Cardy [27]: using modular invariance, the boundary conditions at the end of an interval are mapped into a state in the Hilbert space of the theory defined on a spatial circle. This state is called the *boundary state*.

To this end, we start by working with the theory on a spatial circle. There is a non-chiral $\mathfrak{u}(1)^N$ current algebra, with both holomorphic currents $J_i$ and anti-holomorphic currents $\bar{J}_i$, acting in the obvious way on the $N$ left- and right-moving complex fermions. These are not the currents that we wish to preserve. Instead, the chiral currents are defined by

$$\mathcal{J}_\alpha = Q_{\alpha i} J_i \quad \text{and} \quad \bar{\mathcal{J}}_\alpha = \bar{Q}_{\alpha i} \bar{J}_i \tag{2.4}$$

The boundary state $|\mathcal{R}\rangle$ is defined by the property that no current flows into the boundary. The Sugawara construction then ensures that no energy flows into the boundary either. In terms of the mode expansion of the currents (labelled by $n \in \mathbb{Z}$), this condition reads

$$(\mathcal{J}_{\alpha,n} + \bar{\mathcal{J}}_{\alpha,-n})|\mathcal{R}\rangle = 0 \quad \Rightarrow \quad (\mathcal{R}_{ij} J_{j,n} + \bar{J}_{i,n})|\mathcal{R}\rangle = 0 \tag{2.5}$$

It is not hard to show that solutions to this condition exist if and only if the anomaly constraint (2.2) is satisfied.

The solutions are given in terms of *Ishibashi states* [36]. To define these, first recall the the Hilbert space decomposes into charge sectors under the current algebra generated by $J_i$ and $\bar{J}_i$. In each sector, labelled by its charges $(\lambda_i, \bar{\lambda}_i) \in \mathbb{Z}$, the ground state obeys

$$J_{i,0}|\lambda, \bar{\lambda}\rangle = \lambda_i |\lambda, \bar{\lambda}\rangle \quad \text{and} \quad \bar{J}_{i,0}|\lambda, \bar{\lambda}\rangle = \bar{\lambda}_i |\lambda, \bar{\lambda}\rangle \tag{2.6}$$

These ground states are annihilated by the positive modes, so $J_{i,n}|\lambda, \bar{\lambda}\rangle = J_{i,n}|\lambda, \bar{\lambda}\rangle = 0$ for $n \geq 1$. Excitations above the ground state are then generated by the negative modes, $J_{i,-n}$ and $\bar{J}_{i,-n}$ for $n \geq 1$.

The condition (2.5) must be solved separately in each charge sector. Acting on the ground states, we have

$$\mathcal{R}_{ij}\lambda_j + \bar{\lambda}_i = 0 \tag{2.7}$$

The charge sectors $\lambda_i$ that obey this equation for some choice of $\bar{\lambda}_i$ are precisely those that live in the charge lattice $\Lambda[\mathcal{R}]$ defined in (2.3). Only these charge sectors arise in the boundary state $|\mathcal{R}\rangle$.

In charge sector $\lambda \in \Lambda[\mathcal{R}]$, we can construct the Ishibashi state as the usual coherent sum over excitations [36]. We take $\bar{\lambda} = -\mathcal{R}\lambda$, to obey (2.7) and write

$$\|\lambda, \bar{\lambda}\rangle\rangle = \exp\left(-\sum_{n=1}^\infty \frac{1}{n} \mathcal{R}_{ij} \bar{J}_{i,-n} J_{j,-n}\right) |\lambda, \bar{\lambda}\rangle$$

The boundary state $|\mathcal{R}\rangle$ that we're looking for is then a suitable sum over the Ishibashi states $\|\lambda, \bar{\lambda}\rangle\rangle$ with $\lambda \in \Lambda[\mathcal{R}]$. The coefficients of this sum are fixed by the Cardy-Lewellen sewing conditions [41, 42]. The final result for the boundary state is given by

$$|\theta; \mathcal{R}\rangle = g_{\mathcal{R}} \sum_{\lambda \in \Lambda[\mathcal{R}]} e^{i\gamma_{\mathcal{R}}(\lambda)} e^{i\theta \cdot \lambda} \|\lambda, \bar{\lambda} = -\mathcal{R}\lambda\rangle\rangle \qquad (2.8)$$

There are a number of new ingredients in this expression. The least important is the phase $e^{i\gamma_{\mathcal{R}}(\lambda)}$. An expression for this phase can be found in Appendix B of [35], but it will not play a role in what follows.

More interesting is the phase factor $e^{i\theta \cdot \lambda}$. This arises because there is not a unique solution to the sewing conditions. This means that, for each $\mathcal{R}$, we have a manifold of possible boundary states parameterised by $N$ phases $\theta_i$.

These phases arise even for the simplest boundary conditions, where the reflection of a single left-moving fermion into a right-moving fermion can, in general, be implemented by the boundary condition $\psi_L = e^{i\theta}\psi_R$. The $N$ phases $\theta_i$ that appear in the boundary state (2.8) are generalisation to multiple fermions with a chiral boundary condition $\mathcal{R}$.

**The Central Charge**

The most important new element in (2.8) is the normalisation factor $g_{\mathcal{R}}$. This is determined by insisting that the overlap between any two boundary states can be interpreted, using modular invariance, as the partition function of a sensible theory on the interval. (There is an important caveat in this statement regarding the possible existence of Majorana zero modes; this will be discussed further in Section 4.1.) In [35], we showed that this normalisation factor is given by

$$g_{\mathcal{R}} = \sqrt{\text{Vol}(\Lambda[\mathcal{R}])} \qquad (2.9)$$

Here $\text{Vol}(\Lambda[\mathcal{R}])$ is the volume of the primitive unit cell of the lattice $\Lambda$. This result was previously derived in a somewhat different context in [40].

The normalisation factor is important because it coincides with the Affleck-Ludwig central charge, defined by

$$g_{\mathcal{R}} = \langle 0, 0 \,|\, \theta; \mathcal{R} \rangle$$

Hence, $g_{\mathcal{R}}$ should be thought of as a count of the number of boundary degrees of freedom. This number must strictly decrease in any boundary RG flow.

The trivial boundary conditions, corresponding to $\mathcal{R} = \pm \mathbb{1}$ (or, indeed, to any diagonal $\mathcal{R}$ with entries $\pm 1$.) has $g_{\mathcal{R}} = 1$. This is the smallest value of the central charge. Any chiral boundary conditions necessarily has $g_{\mathcal{R}} > 1$. Each such boundary condition will have a number of relevant operators which induce RG flows. The rest of this paper is concerned with understanding these operators and flows.

## 2.2 Some Examples

With $N = 2$ Dirac fermions, there is a rather simple classification of boundary states. A large class of these arise from taking co-prime integers $(p, q)$ with one odd, one even, and setting

$$Q_{\alpha i} = \begin{pmatrix} p & q \\ -q & p \end{pmatrix} \quad , \quad \bar{Q}_{\alpha i} = \begin{pmatrix} p & -q \\ q & p \end{pmatrix} \quad \Rightarrow \quad \mathcal{R}_{ij} = \frac{1}{c} \begin{pmatrix} a & b \\ -b & a \end{pmatrix} \tag{2.10}$$

Here $a, b$ and $c$ form a Pythogorean triple $a^2 + b^2 = c^2$ with the Euclid parameterisation

$$a = p^2 - q^2 \quad , \quad b = 2pq \quad , \quad c = p^2 + q^2$$

The boundary central charge of these states is simply $g_{\mathcal{R}} = \sqrt{c}$.

The state (2.10) always lies in the vector class of boundary conditions [35]. However, for any choice of central charge, it is not hard to find states that lie in either class. For example, after the trivial states, the simplest states have $g_{\mathcal{R}} = \sqrt{5}$. If we take $p = 2$ and $q = 1$, we get a vector-like boundary state with

$$\mathcal{R}_{ij} = \frac{1}{5} \begin{pmatrix} 3 & 4 \\ -4 & 3 \end{pmatrix}$$

However, flipping the sign of a single row, we get an axial-like boundary state with

$$\mathcal{R}_{ij} = \frac{1}{5} \begin{pmatrix} 3 & 4 \\ 4 & -3 \end{pmatrix}$$

As we proceed, many of the key ideas will be illustrated by this $g_{\mathcal{R}} = \sqrt{5}$ state. For now, there are a couple of points worth highlighting.

First, the fact that sign-flipping a row or column of $\mathcal{R}$ changes the topological class is a property of all boundary states. Meanwhile, permuting rows or columns leaves the class unchanged. In general, one can transform $\mathcal{R} \rightarrow P_R \mathcal{R} P_L$ where $P_L$ and $P_R$ are signed permutation matrices. This transformation corresponds to acting with a Weyl group element $(W_L, W_R) \in O(2N)_L \times O(2N)_R$ on the boundary state; the class then changes if $\det(W_L)\det(W_R) = -1$, while $g_{\mathcal{R}}$ always stays the same. This illustrates the fact that, for any given choice of $g_{\mathcal{R}}$, there are boundary states that lie in both classes.

Secondly, a number of different charges $Q$ and $\bar{Q}$ share the same boundary state, characterised by $\mathcal{R}$. For example, we could also take

$$Q_{\alpha i} = \begin{pmatrix} 3 & 4 \\ -4 & 3 \end{pmatrix} \quad , \quad \bar{Q}_{\alpha i} = \begin{pmatrix} 5 & 0 \\ 0 & 5 \end{pmatrix} \quad \Rightarrow \quad \mathcal{R}_{ij} = \frac{1}{5} \begin{pmatrix} 3 & 4 \\ -4 & 3 \end{pmatrix}$$

In contrast to the charge matrices in (2.10), here the $U(1)^2$ symmetry does not act faithfully on the bulk fermions. The fermions are untouched by a discrete $\mathbb{Z}_5$ which acts on the left-movers as $\psi_i \to e^{i\beta_\alpha Q_{\alpha i}}\psi_i$ and on the right-movers as $\bar{\psi}_i \to e^{i\beta_\alpha \bar{Q}_{\alpha i}}\bar{\psi}_i$, with $\beta = (\frac{2\pi}{5}, \frac{4\pi}{5})$.

In what follows, the key physics will depend only on $\mathcal{R}$; for example, the collection of relevant boundary operators and their dimensions depend only on $\mathcal{R}$. Nonetheless, we will see that the charges of these operators are inherited from $Q$ and $\bar{Q}$ and so require extra information beyond a knowledge of $\mathcal{R}$.

**Another Example: the Maldacena-Ludwig state**

Our second example involves $N = 4$ Dirac fermions. The boundary conditions are, perhaps, most simply described by requiring an $SU(4) \times U(1)$ global symmetry under which the left-movers transform in the $\mathbf{4}_{+1}$ representation, while the right-movers transform as $\mathbf{4}_{-1}$. There is no linear boundary condition on the fermions that reflects one into another, a fact first noted in the context of monopole physics [43, 44]. Instead, the boundary condition is implemented by the boundary state with

$$Q_{\alpha i} = \begin{pmatrix} + & + & + & + \\ + & - & & \\ & & + & - \\ & & & + & - \end{pmatrix} \quad , \quad \bar{Q}_{\alpha i} = \begin{pmatrix} - & - & - & - \\ + & - & & \\ & & + & - \\ & & & + & - \end{pmatrix} \quad \Rightarrow \quad \mathcal{R}_{ij} = \delta_{ij} - \frac{1}{2} \tag{2.11}$$

This boundary state was previously introduced by Maldacena and Ludwig [34]. It manifestly implements the symmetry of the Cartan subalgebra $U(1)^4 \subset SU(4) \times U(1)$. Less manifestly, it also preserves the full $SU(4) \times U(1)$. Remarkably, in this special four-fermion case, it preserves yet a larger $SO(8)/\mathbb{Z}_2$ symmetry group, whose existence can be traced to triality. This state has boundary central charge $g_\mathcal{R} = \sqrt{2}$. Once again, by acting with Weyl group transformations we have such states of either $\mathbb{Z}_2$ SPT class.

The Maldacena-Ludwig state also has a somewhat different avatar: it is the state that implements the Fidkowski-Kitaev gapped phase of 8 Majorana fermions, an interpretation that was first made in [28].

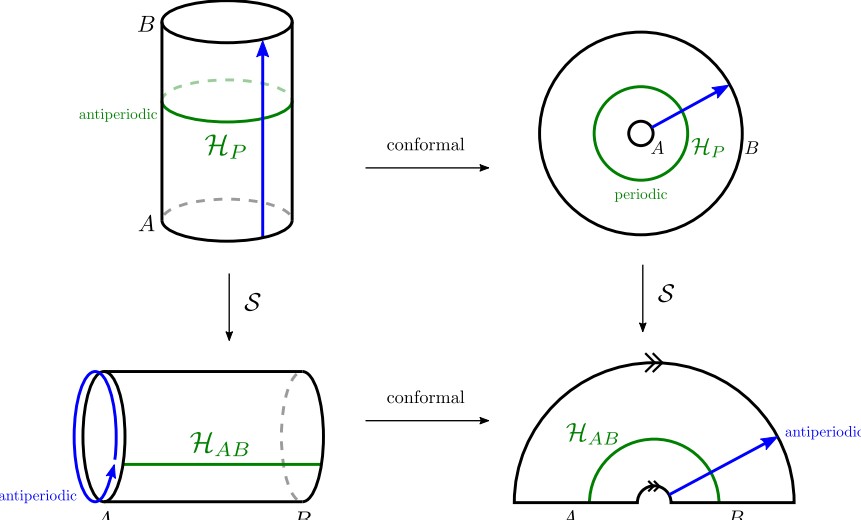

**Figure 1**: The various conformal identifications used, including those which correspond to an $\mathcal{S}$ transformation of the argument of the partition function.

## 3   The Partition Function

Our goal in this section is to determine the relevant boundary operators, and their charges, for each choice of boundary condition $\mathcal{R}$. To do this, we compute the partition function of the theory on an interval, with boundary conditions $\mathcal{R}$ imposed on each end. This encodes the information about the states in the Hilbert space on the interval. We then use the state-operator map to determine the spectrum of boundary operators.

The partition function $\mathcal{Z}_{AB}$, for two distinct boundary conditions $A$ and $B$ at either end of the interval is defined as the trace over the Hilbert space, $\mathcal{H}_{AB}$. After implementing a conformal transformation to the half-annulus, as shown in Figure 1 along the bottom row, this partition function is given by

$$\mathcal{Z}_{AB}(q) = \text{Tr}_{\mathcal{H}_{AB}}\left(q^{L_0 - c/24}\right)$$

In the presence of a boundary, only one copy of the Virasoro generators survives. These we label as $L_n$, though they are distinct from the bulk holomorphic generators. The usual Cardy trick is to relate this "open string" partition function to the "closed string" partition function of free fermions on a cylinder which, after the conformal map shown along the top of Figure 1, becomes the annulus

$$\mathcal{Z}_{\text{closed}}(q) = \langle B|q^{\frac{1}{2}(L_0 + \bar{L}_0 - c/12)}|A\rangle$$

which now includes contributions from both holomorphic $L_0$ and anti-holomorphic $\bar{L}_0$ generators. The fermions are given periodic boundary conditions on the annulus (inherited, in the usual way, from anti-periodic boundary conditions on the cylinder before the conformal map.) The open and closed string partition functions are then related by a modular $\mathcal{S}$-transformation of $q$.

Consider two boundary states $A = |\theta, \mathcal{R}\rangle$ and $B = |\theta', \mathcal{R}\rangle$ of the form (2.8). Note that these states share the same $\mathcal{R}$ matrix, but differ in the theta angles. The general closed string partition function was computed in [35]; it is

$$\mathcal{Z}_{\text{closed}}(q) = g_{\mathcal{R}}^2 \sum_{\lambda \in \Lambda[\mathcal{R}]} e^{i(\theta - \theta' + \pi) \cdot \lambda} \frac{q^{\lambda^2/2}}{\eta(\tau)^N} \tag{3.1}$$

Here $q = e^{2\pi i \tau}$. The slightly unusual factor of $e^{i\pi \cdot \lambda} := e^{i\pi(\lambda_1 + \cdots + \lambda_N)}$ arises from an insertion of holomorphic fermion parity $(-1)^F = (-1)^{\lambda_1 + \cdots + \lambda_N}$, whose necessity was pointed out in [29]. The partition function for the theory on the interval is then found by applying a modular $\mathcal{S}$-transformation; it is

$$\mathcal{Z}_{AB}(q) = \sum_{\rho \in \Lambda[\mathcal{R}]^\star} \frac{q^{\frac{1}{2}(\rho + \frac{\theta - \theta'}{2\pi} + \frac{1}{2})^2}}{\eta(\tau)^N} \tag{3.2}$$

with $\Lambda[\mathcal{R}]^\star$ the dual lattice, defined by $\rho \cdot \lambda \in \mathbb{Z}$ for all $\lambda \in \Lambda[\mathcal{R}]$ and $\rho \in \Lambda[\mathcal{R}]^\star$.

## 3.1 Adding Fugacities

Here we wish to extend this computation to include fugacities for the $U(1)^N$ symmetry, providing information about the charges of the states. This means that we weight the contribution of states in the open-string partition function according to their charges under

$$\mathcal{Q}_\alpha = \frac{1}{2\pi i} \int_C dz \, \mathcal{J}_\alpha(z) - d\bar{z} \, \bar{\mathcal{J}}_\alpha(\bar{z})$$

where the contour $C$ is the counter-clockwise semi-circle shown in Figure 2a. The partition function now depends both on the modular parameter $q$ and the chemical potentials $\mu_\alpha$,

$$\mathcal{Z}_{AB}(q; \mu) = \text{Tr}_{\mathcal{H}_{AB}} \left( q^{L_0 - c/24} e^{i\mu_\alpha \mathcal{Q}_\alpha} \right)$$

Again, this object is simplest to compute in the closed-string picture. The operator $e^{i\mu_\alpha \mathcal{Q}_\alpha}$ is now a defect, oriented along the "temporal" or "thermal" direction, as shown in Figure 2b. Its role is to shift each fermion by a phase as we move around the spatial circle. The left-moving fermion $\psi_i$ picks up a phase $e^{i\mu_\alpha Q_{\alpha i}}$, while the right-moving fermion $\bar{\psi}_i$ picks up $e^{i\mu_\alpha \bar{Q}_{\alpha i}}$.

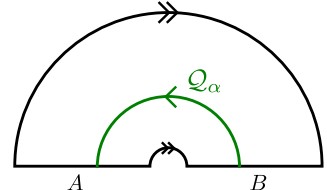

(a) Contour $C$ used to define $\mathcal{Q}_\alpha$.

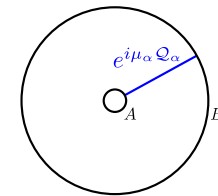

(b) Corresponding defect operator.

**Figure 2**

This, in turn, affects the quantisation of the charges $\lambda_i$ and $\bar{\lambda}_i$ defined in (2.6). Rather than living in the integer lattice $\mathbb{Z}^N$, we instead have

$$\lambda_i \in \mathbb{Z} + \frac{\mu_\alpha Q_{\alpha i}}{2\pi} \quad \text{and} \quad \bar{\lambda}_i \in \mathbb{Z} - \frac{\mu_\alpha \bar{Q}_{\alpha i}}{2\pi} \tag{3.3}$$

Note that left- and right-moving charges are shifted in opposite directions. (This computation leaves an ambiguity in the overall sign of the shifts, which is unimportant for what follows.)

The boundary condition (2.5) still requires that left- and right-moving charges are related by (2.7)

$$Q_{\alpha i}\lambda_i + \bar{Q}_{\alpha i}\bar{\lambda}_i = 0$$

which is only possible for all choices of $\mu$ if

$$\mu_\beta(Q_{\alpha i}Q_{\beta i} - \bar{Q}_{\alpha i}\bar{Q}_{\beta i}) = 0$$

Happily this follows from the condition for vanishing 't Hooft anomalies (2.2).

The closed string partition function is now easily computed by implementing the shift (3.3) in our previous result (3.1). The contribution from the $e^{i(\theta-\theta'+\pi)\cdot\lambda}$ term gives an overall phase which we ignore. We're then left with

$$\mathcal{Z}_{\text{closed}}(q;\mu) = g_{\mathcal{R}}^2 \sum_{\lambda \in \Lambda[\mathcal{R}]} e^{i(\theta-\theta'+\pi)\cdot\lambda} \frac{q^{\frac{1}{2}(\lambda_i + \mu_\alpha Q_{\alpha i}/2\pi)^2}}{\eta(\tau)^N}$$

We can now invoke the usual modular transformation to compute the open-string partition function of interest. We pull back the function $Z_{\text{closed}}$ under a modular S-transformation of $q$, to find

$$\mathcal{Z}_{AB}(q;\mu) = \text{Vol}(\Lambda[\mathcal{R}]) \int d^N x \, e^{i\mu_\alpha Q_{\alpha i}x_i} \frac{q^{x^2/2}}{\eta(\tau)^N} \sum_{\lambda \in \Lambda[\mathcal{R}]} e^{i(\theta-\theta'+\pi+2\pi x)\cdot\lambda}$$

Upon doing the integral, we have

$$\mathcal{Z}_{AB}(q;\mu) = \sum_{\rho \in \Lambda[\mathcal{R};\Delta\theta]^\star} e^{i\mu^T Q\rho} \frac{q^{\rho^2/2}}{\eta(\tau)^N} \tag{3.4}$$

The difference from our previous result (3.2) lies in both the explicit $\mu$ dependent factor $e^{i\mu_\alpha Q_{\alpha i}\rho_i}$, and in the sum which now runs over the shifted dual lattice

$$\Lambda[\mathcal{R};\Delta\theta]^\star := \Lambda[\mathcal{R}]^\star + \frac{\theta' - \theta + \pi}{2\pi}$$

The highest weight states are labelled by vectors $\rho \in \Lambda[\mathcal{R};\Delta\theta]^\star$. From (3.4), we can read off their charges

$$\mathcal{Q}_\alpha = Q_{\alpha i}\rho_i \tag{3.5}$$

and energy

$$L_0 = \frac{1}{2}\rho^2 = \frac{1}{2}\mathcal{Q}_\alpha \, \mathcal{M}_{\alpha\beta}^{-1} \, \mathcal{Q}_\beta \tag{3.6}$$

where we have introduced the matrix $\mathcal{M}_{\alpha\beta} = Q_{\alpha i}Q_{\beta i} = \bar{Q}_{\alpha i}\bar{Q}_{\beta i}$. This latter equality, relating the charges to the energy, is consistent with the Sugawara construction.

### 3.2 Boundary Operators

The state-operator map means that the partition function also contains information about the spectrum of boundary operators. To extract this information, we set $\theta = \theta'$ and drop the contribution of $\pi$ from the $(-1)^F$ factor. The boundary operators are then labelled by $\rho \in \Lambda[\mathcal{R}]^\star$. Like the states, the operators have charges $\mathcal{Q}_\alpha$ and dimension $L_0$, again given by (3.5) and (3.6).

Boundary operators also come in one of two classes: they are fermionic or bosonic. This fermion parity will play a key role in Section 4 where we discuss RG flows initiated by such operators. We pause here to discuss how to classify operators. As we now explain, it is possible to assign a fermion parity to the lattice vectors $\rho \in \Lambda[\mathcal{R}]^\star$.

First, recall that by definition, under a $U(1)_L^N \times U(1)_R^N$ transformation

$$(e^{i\mu_\alpha Q_{\alpha i}}, e^{i\mu_\alpha \bar{Q}_{\alpha i}})$$

belonging to the preserved $U(1)^N$ subgroup, the boundary operator labelled by $\rho$ picks up a phase $e^{i\mu_\alpha \mathcal{Q}_\alpha} = e^{i\mu_\alpha Q_{\alpha i}\rho_i}$. Importantly, the bulk fermion parity operator $(-1)^{F+\bar{F}}$

is of the above form [32]. That is, there exists a choice of $\mu_\alpha$ for which the above transformation is

$$(e^{i\mu_\alpha Q_{\alpha i}}, e^{i\mu_\alpha \bar{Q}_{\alpha i}}) = (-1, \ldots, -1, -1, \ldots, -1)$$

It will be more convenient to work not with $\mu_\alpha$, but with the vector $f_i = \mu_\alpha Q_{\alpha i}/\pi$. We shall refer to this as the "fermion vector". The above condition can then be written

$$(e^{i\pi f}, e^{i\pi \mathcal{R}f}) = (-1, \ldots, -1, -1, \ldots, -1)$$

which shows that $f$ is characterised by the requirement that both $f$ and $\mathcal{R}f$ are odd-integer vectors. It therefore naturally lives in $\Lambda[\mathcal{R}]/2\Lambda[\mathcal{R}]$. With this notation in hand, the key point is then that since fermion parity lies within $U(1)^N$, the charge $\rho$ dictates the fermion parity $(-1)^F$ of the boundary operator[3], through

$$(-1)^F = e^{i\mu_\alpha Q_{\alpha i}\rho_i} = (-1)^{f\cdot\rho} \tag{3.7}$$

We therefore classify vectors $\rho \in \Lambda[\mathcal{R}]^\star$ as bosonic or fermionic depending on whether $\rho \cdot f$ is even or odd, respectively.

Relevant boundary operators are associated to lattice vectors $\rho \in \Lambda[\mathcal{R}]^\star$ with $\rho^2 < 2$ and can be either bosonic or fermionic. These will be our primary focus in Section 4 where we discuss RG flows initiated by such operators. Here we describe the relevant operators in the two examples introduced in Section 2.2.

### The First Example: $g = \sqrt{5}$

As we've seen, the simplest, non-trivial two fermion boundary state has

$$\mathcal{R}_{ij} = \begin{pmatrix} 3/5 & 4/5 \\ -4/5 & 3/5 \end{pmatrix}$$

and $g_\mathcal{R} = \sqrt{5}$. One possible choice of the fermion vector in this case is $f = (5, 5)$.

As we explained in Section 2.2, there are many choices of $Q_{\alpha i}$ and $\bar{Q}_{\alpha i}$ that give rise to this boundary state. The dimension of boundary operators depends only on $\mathcal{R}_{ij}$ while, as we see from (3.5), the charges of these operators depend on the choice of $\mathcal{Q}$. The operators are further distinguished by fermion number $(-1)^F$. The operators with $L_0 \leq 1$ are associated to the following lattice sites $\rho$,

---

[3]Just as for the Virasoro generators $L_n$, the notation $(-1)^F$ is ambiguous, and means something different depending on whether one is working in the open or closed sector.

| $L_0$ | $(-1)^F$ | $\rho \in \Lambda[\mathcal{R}]^\star$ |
|---|---|---|
| $0$ | $+$ | $(0,0)$ |
| $1/10$ | $-$ | $\pm(\frac{2}{5},\frac{1}{5}),\ \pm(\frac{1}{5},-\frac{2}{5})$ |
| $1/5$ | $+$ | $\pm(\frac{1}{5},\frac{3}{5}),\ \pm(\frac{3}{5},-\frac{1}{5})$ |
| $2/5$ | $+$ | $\pm(\frac{4}{5},\frac{2}{5}),\ \pm(\frac{2}{5},-\frac{4}{5})$ |
| $1/2$ | $-$ | $\pm(\frac{3}{5},\frac{4}{5}),\ \pm(\frac{4}{5},-\frac{3}{5}),\ \pm(1,0),\ \pm(0,1)$ |
| $4/5$ | $+$ | $\pm(\frac{2}{5},\frac{6}{5}),\ \pm(\frac{6}{5},-\frac{2}{5})$ |
| $9/10$ | $-$ | $\pm(\frac{6}{5},\frac{3}{5}),\ \pm(\frac{3}{5},-\frac{6}{5})$ |
| $1$ | $+$ | $\pm(\frac{7}{5},\frac{1}{5}),\ \pm(\frac{1}{5},-\frac{7}{5}),\ \pm(1,1),\ \pm(1,-1)$ |

As we proceed, we'll see the interpretation of a number of these operators.

### The Other Example: The Maldacena-Ludwig State

The relevant boundary operators for the Maldacena-Ludwig state (2.11) are

| $L_0$ | $(-1)^F$ | $\rho \in \Lambda[\mathcal{R}]^\star$ | |
|---|---|---|---|
| $0$ | $+$ | $(0,0)$ | |
| $1/2$ | $+$ | $\pm(\frac{1}{2},\frac{1}{2},\frac{1}{2},\frac{1}{2}),\ (\frac{1}{2},\frac{1}{2},-\frac{1}{2},-\frac{1}{2})$ | (and all permutations) |
| $1/2$ | $-$ | $(\pm1,0,0,0),\ \pm(\frac{1}{2},\frac{1}{2},\frac{1}{2},-\frac{1}{2})$ | (and all permutations) |
| $1$ | $+$ | $(\pm1,\pm1,0,0)$ | (and all permutations) |

As we briefly mentioned previously, the Maldacena-Ludwig state represents the gapped Fidkowski-Kitaev state. This has the property that it preserves both left and right fermion parity $(-1)^F$ and $(-1)^{\bar{F}}$. Furthermore, it is the state with the smallest $g_\mathcal{R}$ with this property. This latter statement is reflected in the fact that the dimension $L_0 = \frac{1}{2}$ bosonic operators are charged under both of the two fermionic parities. We will return to these aspects of the boundary states in [32].

### Marginal Operators

Marginal boundary operators have $L_0 = 1$. If such operators are exactly marginal, they give rise to continuous families of boundary states. As we now explain, marginal operators fall into a number of different categories.

First, we can take the vacuum module, $\rho = 0$, and form a level-1 descendent under the current algebra. From the perspective of the interval Hilbert space, these correspond to states $\mathcal{J}_{\alpha,-1}|0\rangle$. Similarly, the existence of the boundary operators follows on symmetry grounds: they are associated to the symmetries broken by the boundary in the reduction $U(1)_L^N \times U(1)_R^N \to U(1)^N$. Acting with these operators changes the $\theta$-angles that, as we saw in (2.8), are needed to characterise the boundary state.

The second class of marginal operators are highest weight states associated to lattice vectors $\rho \in \Lambda[\mathcal{R}]^\star$ with $\rho^2 = 2$. We have listed these operators in the table above for the simple examples. Many of these operators also have an interpretation in terms of symmetries. But not all.

To understand this, first recall that the symmetry breaking pattern, as shown in (2.1), is generically

$$\mathfrak{so}(2N)_L \times \mathfrak{so}(2N)_R \to \mathfrak{u}(1)^N$$

The broken, off-diagonal elements of $\mathfrak{so}(2N)_L \times \mathfrak{so}(2N)_R$ will also give rise to marginal operators. Acting with them simply rotates the unbroken Cartan sub-algebra.

It is straightforward to identify these states. The off-diagonal elements of $so(2N)_L$ arise from vectors with $\rho^2 = 2$ that sit in $\rho \in \mathbb{Z}^N$. The off-diagonal elements of $so(2N)_R$ arise from vectors with $\rho^2 = 2$ that sit in $\rho \in \mathcal{R}^{-1}\mathbb{Z}^N$.

This pattern can be clearly seen in the two fermion boundary state with $g_\mathcal{R} = \sqrt{5}$. The final line of the table shows the 8 boundary operators that are associated to the off-diagonal elements of $SO(4)_L \times SO(4)_R$.

However, in other examples things may not be so straightforward. First, it may be that there is an overlap between the operators associated to $\mathfrak{so}(2N)_L$ and those associated to $\mathfrak{so}(2N)_R$. This occurs if there are lattice sites with $\rho^2 = 2$ that sit in $\rho \in \mathbb{Z}^N \cap \mathcal{R}^{-1}\mathbb{Z}^N$. But the intersection of the latter two lattices is simply

$$\Lambda[\mathcal{R}] = \mathbb{Z}^N \cap \mathcal{R}^{-1}\mathbb{Z}^N$$

This overlap has a very natural interpretation. As explained in [32], vectors $\rho \in \Lambda[\mathcal{R}]$ with $\rho^2 = 2$ correspond to enhanced symmetries of the boundary state. As expected, the presence of such hidden symmetries reduces the number of marginal boundary operators. For example, in the table for the Maldacena-Ludwig boundary state shown above, there are 24 marginal operators. This is lower than the number 48 of off-diagonal generators of $SO(8)_L \times SO(8)_R$. The difference can be accounted for by the enhanced $SO(8)/\mathbb{Z}_2$ symmetry, which eliminates 24 generators.

Finally, some boundary states have marginal operators that do not correspond to symmetries. These are lattice vectors with $\rho^2 = 2$ that sit in $\rho \in \Lambda[\mathcal{R}]^\star$ but with $\rho \notin \mathbb{Z}^N \cup \mathcal{R}^{-1}\mathbb{Z}^N$. In such cases, one must work harder to determine whether the the boundary operator is exactly marginal, or marginally relevant or irrelevant. We will not explore this issue further.

### 3.3 An Aside: The Unitarity "Paradox"

There is an interesting structure to the charges carried by states in the Hilbert space $\mathcal{H}_{AB}$. To illustrate our point, it's simplest if we ignore the shift of the lattice by the theta angles for now, so $\rho \in \Lambda[\mathcal{R}]^\star$. In this case, the states of the Hilbert space carry charges in the lattice (3.5)

$$\mathcal{Q} \in Q\Lambda[\mathcal{R}]^\star$$

We can compare this to the charges of states that we get by acting with left- and right-moving operators. Acting with the holomorphic fermions $\psi_i$ produce states with charges in $Q\mathbb{Z}^N$, while acting with anti-holomorphic fermions $\bar{\psi}_i$ produce states with charges in $\bar{Q}\mathbb{Z}^N$. It is not hard to show that this accounts for the full charge lattice

$$Q\Lambda[\mathcal{R}]^\star = Q\mathbb{Z}^N + \bar{Q}\mathbb{Z}^N$$

However, there's a twist. It's not true that one can reach states of all charges by acting only with, say, holomorphic operators. This is, at heart, what it means for our boundary states to be chiral. Indeed, we have the following:

$$\left[Q\Lambda[\mathcal{R}]^\star : Q\mathbb{Z}^N\right] = \left[Q\Lambda[\mathcal{R}]^\star : \bar{Q}\mathbb{Z}^N\right] = \mathrm{Vol}(\Lambda[\mathcal{R}])$$

This means that, while one cannot access states of any charge by acting on the vacuum with only holomorphic operators, we can do so by acting on an appropriate choice of $g_\mathcal{R}^2 = \mathrm{Vol}(\Lambda[\mathcal{R}])$ states (one of which is the ground state). These can be viewed as holomorphic superselection sectors.

Similarly, there are a different set of $g_\mathcal{R}^2$ states in the Hilbert space, from which we can access states of any charge by acting with anti-holomorphic operators.

In the context of scattering off a single boundary, this leads to a seeming "unitarity paradox". It is not hard to set up situations in which a single left-moving fermion scatters off the boundary, but cannot return as any combination of right-moving fermions. This is captured by the vanishing correlation functions

$$\langle 0|\psi_i(z)\bar{\psi}_{j_1}(\bar{z}_1)\ldots\bar{\psi}_{j_N}(\bar{z}_N)|0\rangle = 0 \quad \text{for all } N$$

Such behaviour was seen, for example, in [33, 34, 43, 44]. Our general discussion above shows that the right-moving fermions are not excitations above the ground state, but instead above one of the other $\mathrm{Vol}(\Lambda[\mathcal{R}])$ superselection sectors.

# 4   RG Flows: Statements

We now turn to the main results of this paper. We will follow the RG flow between different boundary states.

We start with a given UV boundary state, preserving the $U(1)^N$ symmetry characterised by the charge matrix $\mathcal{R}_{UV}$. As we have seen, relevant boundary operators are labelled by a vector $\rho \in \Lambda[\mathcal{R}_{UV}]^\star$ and carry charge

$$\mathcal{Q}_\alpha = Q_{\alpha i}\rho_i$$

We turn on a single, relevant, bosonic boundary operator of definite charge to initiate an RG flow. Along the flow, the symmetry is broken to

$$U(1)^N \to U(1)^{N-1}$$

In what follows, we make the following, important assumption: *At the end of the flow, an emergent $U(1)^N$ symmetry is again restored.* This means that, in the infra-red, the physics is again described by a boundary state of the form (2.8), now with a different charge matrix $\mathcal{R}_{IR}$.

There is, in fact, a unique choice for $\mathcal{R}_{IR}$ for each relevant operator labelled by $\rho$. This follows because of the $U(1)^{N-1}$ symmetry that exists along the RG flow. This symmetry must be preserved by the IR boundary state, a condition which translates into the simple requirement that

$$\mathcal{R}_{IR}\bigg|_{\rho^\perp} = \mathcal{R}_{UV}\bigg|_{\rho^\perp} \tag{4.1}$$

or in other words, that the two matrices must agree on the orthogonal complement of $\rho$. But for orthogonal matrices, this condition is highly constraining. In particular, there are only two options for $\mathcal{R}_{IR}$. One is $\mathcal{R}_{UV}$ itself, but this is quickly ruled out by the fact that $g_{IR} = g_{UV}$, in contravention of the $g$-theorem which states that the central charge must strictly decrease under relevant perturbations. This only leaves the second option, which is that the matrices differ by a reflection along the vector $\rho$:

$$(\mathcal{R}_{IR})_{ik} = (\mathcal{R}_{UV})_{ij}\left(\delta_{jk} - \frac{2}{\rho^2}\rho_j\rho_k\right) \tag{4.2}$$

The second factor is the matrix implementing the reflection along $\rho$.

One might think that the infra-red central charge is, following (2.9),

$$g_{\text{naive}} = \sqrt{\text{Vol}(\Lambda[\mathcal{R}_{IR}])} \tag{4.3}$$

And, for some of the RG flows, where no subtleties arise, this indeed the correct answer. However, it is not true in general. There are two rather interesting effects that may occur, both of which leave us with an infra-red central charge larger than (4.3). First, certain RG flows necessarily result in a Majorana zero mode stuck on the boundary. This phenomenon, which is explained in Section 4.1, increases the normalisation of the boundary state and its central charge by a factor of $\sqrt{2}$. Secondly, some RG flows result in a superposition of primitive boundary states, and larger central charge. This phenomenon is explained in 4.2.

Furthermore, we will see that the infra-red central always obeys the $g$-theorem, which states that the boundary central charge must always decrease [1–3]. This fact arises in a mathematically non-trivial manner for our boundary states, and presents a stringent test of the assumption a full $U(1)^N$ symmetry emerges in the infra-red.

## 4.1 Majorana Zero Modes

As we explained in the introduction, boundary conditions fall into two distinct topological classes, characterised by a mod 2 anomaly. One might have thought that RG flows would remain within a given class. However, as we now describe, our conjecture (4.2) does *not* have this property. It is not difficult to find RG flows that go from one class to another, and we present examples below. We will explain why this is not problematic.

First, we review the result of [35] that determines the topological class in which a given boundary state, labelled by $\mathcal{R}$, sits. Given a CFT on an interval, we can impose different boundary conditions $\mathcal{R}$ and $\mathcal{R}'$ on either end. In [35], we derived a simple formula for the number of ground states $G[\mathcal{R}, \mathcal{R}']$ of this system:

$$G[\mathcal{R}, \mathcal{R}'] = \frac{\sqrt{\text{Vol}(\Lambda[\mathcal{R}]) \, \text{Vol}(\Lambda[\mathcal{R}'])}}{\text{Vol}(\Lambda[\mathcal{R}, \mathcal{R}'])} \sqrt{\det'(\mathbb{1} - \mathcal{R}^T \mathcal{R}')} \tag{4.4}$$

Here the intersection lattice $\Lambda[\mathcal{R}, \mathcal{R}']$ is defined to be those integer vectors $\lambda$ for which $\mathcal{R}\lambda = \mathcal{R}'\lambda \in \mathbb{Z}^N$. The notation $\det'$ denotes the product over non-vanishing eigenvalues.

The ground state degeneracy has an interesting property. If the two boundary states $\mathcal{R}$ and $\mathcal{R}'$ lie in the same class (i.e. either both vector, or both axial) then the number of ground states is integer as expected

$$G[\mathcal{R}, \mathcal{R}'] \in \mathbb{Z}$$

In contrast, if the two boundary states lie in different classes, then

$$G[\mathcal{R}, \mathcal{R}'] \in \sqrt{2}\,\mathbb{Z}$$

The $\sqrt{2}$ factor reflects the existence of a bulk Majorana zero mode. This is telling us that it is not consistent to put boundary conditions from different classes at the two ends of an interval. A discussion of which class a general boundary condition $\mathcal{R}$ sits in can be found in [35].

What to make of the fact that RG flows take us from one class to another? Clearly, a consistent quantum system, with compatible boundary conditions on each end, cannot flow to an inconsistent quantum system. It must be that the bulk Majorana mode that appears in the infra-red is accompanied by a second, boundary Majorana mode. This boundary Majorana mode contributes a further factor of $\sqrt{2}$ to the partition function, as in (1.1), and hence to the boundary central charge. This means that, if there's no further subtlety, RG flows which interpolate between different classes have

$$g_{IR} = \sqrt{2\,\text{Vol}(\Lambda[\mathcal{R}_{IR}])} \tag{4.5}$$

The condition for the appearance of a boundary Majorana mode is encoded in a simple property of $\rho$. First, we recall that although $\rho \in \Lambda[\mathcal{R}_{UV}]^\star$, it need not be *primitive* within this lattice. Instead, it may be possible to write it as some multiple $n \geq 1$ of an underlying primitive vector, which we denote as $\hat{\rho}$:

$$\rho = n\hat{\rho} \tag{4.6}$$

Since we must perturb by a bosonic relevant operator, $\rho$ is always required to be bosonic. But there is no such condition on $\hat{\rho}$. In particular, it is perfectly acceptable for $\hat{\rho}$ to be fermionic provided that $n$ is even. The property of $\rho$ which determines the existence of a boundary mode is then the fermionic/bosonic nature of $\hat{\rho}$. This follows by computing the ground state degeneracy (4.4) between $\mathcal{R}_{IR}$ and $\mathcal{R}_{UV}$; as we show in Section 5, is given by

$$G[\mathcal{R}_{UV}, \mathcal{R}_{IR}] = \begin{cases} 1 & \text{if } \hat{\rho} \text{ is bosonic} \\ \sqrt{2} & \text{if } \hat{\rho} \text{ is fermionic} \end{cases} \tag{4.7}$$

In other words, there is a bulk Majorana zero mode only if the relevant operator is associated to a lattice vector $\rho = n\hat{\rho}$ built on a fermionic primitive vector $\hat{\rho}$.

In Appendix B, we give more details illustrating the coupling between the boundary mode and the bulk fermions using a simple model.

**An Example**

We can illustrate these ideas with the example that we met in Section 2.2: two fermions with

$$\mathcal{R}_{UV} = \begin{pmatrix} 3/5 & 4/5 \\ -4/5 & 3/5 \end{pmatrix}$$

The boundary central charge is $g_{UV} = \sqrt{5}$.

We listed the relevant and marginal operators for this boundary state in Section 3.2. Here we are interested only in the relevant, bosonic operators. For each of these, we can determine the infra-red charge matrix and whether or not there exists a boundary Majorana zero mode at the end of the flow.

| $\rho$ | $L_0$ | $\mathcal{R}_{IR}$ | Majorana? |
|---|---|---|---|
| $\left(\frac{1}{5}, \frac{3}{5}\right)$ | $\frac{1}{5}$ | $\begin{pmatrix} 0 & -1 \\ -1 & 0 \end{pmatrix}$ | No |
| $\left(\frac{3}{5}, -\frac{1}{5}\right)$ | $\frac{1}{5}$ | $\begin{pmatrix} 0 & 1 \\ 1 & 0 \end{pmatrix}$ | No |
| $\left(\frac{4}{5}, \frac{2}{5}\right)$ | $\frac{2}{5}$ | $\begin{pmatrix} -1 & 0 \\ 0 & 1 \end{pmatrix}$ | Yes |
| $\left(\frac{2}{5}, -\frac{4}{5}\right)$ | $\frac{2}{5}$ | $\begin{pmatrix} 1 & 0 \\ 0 & -1 \end{pmatrix}$ | Yes |
| $\left(\frac{2}{5}, \frac{6}{5}\right)$ | $\frac{4}{5}$ | $\begin{pmatrix} 0 & -1 \\ -1 & 0 \end{pmatrix}$ | No |
| $\left(\frac{6}{5}, -\frac{2}{5}\right)$ | $\frac{4}{5}$ | $\begin{pmatrix} 0 & 1 \\ 1 & 0 \end{pmatrix}$ | No |

The middle two rows are built on the underlying fermionic vectors $\pm\left(\frac{2}{5}, \frac{1}{5}\right)$ and $\pm\left(\frac{1}{5}, -\frac{2}{5}\right)$, while the remaining rows are built on bosonic vectors. Note that the $\rho$-vectors for the operators with dimension $\frac{4}{5}$ are proportional to those with dimension $\frac{1}{5}$. We'll see the difference between these two RG flows in the next section.

An analogous table, for a more complicated example, is given in Appendix A.

**Flows with Fermionic Operators**

RG flows are always initiated by bosonic, relevant operators. As we've seen, at the end of an RG flow we may end up with a localised Majorana fermion. We could also ask: what happens if we start from a boundary condition with such a Majorana mode?

The boundary state including such a Majorana mode is simply given by[4] $\sqrt{2}\,|\theta;\mathcal{R}_{UV}\rangle$, and has central charge

$$g_{UV} = \sqrt{2\,\mathrm{Vol}(\Lambda[\mathcal{R}_{UV}])}$$

Starting with such a state opens up a new possibility, because we could dress boundary fermionic operators with the Majorana mode to give a bosonic boundary operator, and then use this to initiate the RG flow.

Such fermionic boundary operators are characterised by $\rho = n\hat{\rho}$, as in (4.6), with $\hat{\rho}$ fermionic, $n$ odd. Because $\hat{\rho}$ is fermionic, this means that such flows always flip the SPT class, and the Majorana mode is absorbed along the flow. The absorption of the Majorana mode means that the infra-red central charge is reduced by an extra factor of $\sqrt{2}$.

The Maldacena-Ludwig state serves as a good example of fermionic flows. Recall that this state has boundary central charge $g = \sqrt{2}$. If we further add a Majorana mode, the central charge is $g_{UV} = 2$. We can now perturb this state by relevant fermionic operators.

These operators were listed in the table in Section 3.2: there are two kinds, with charge given by permutations of

$$\rho = (1,0,0,0) \quad \text{and} \quad \rho = (\tfrac{1}{2}, \tfrac{1}{2}, \tfrac{1}{2}, -\tfrac{1}{2})$$

These are primitive vectors, and both have dimension $L_0 = \tfrac{1}{2}$. Deforming by any of these operators gives us back the Maldacena-Ludwig state, up to a Weyl group transformation of $O(8)_L \times O(8)_R$. In other words, the sole effect of the flow is to eliminate the Majorana mode from the boundary.

In fact, this kind of flow, in which the Majorana is killed (presumably on a boat) is possible for all boundary states. All such states have a boundary fermionic operator of dimension $\tfrac{1}{2}$, since this is simply the bulk fermion brought to the boundary. Deforming by this operator initiates an RG flow from $\sqrt{2}\,|\mathcal{R}\rangle$ to $|\mathcal{R}'\rangle$, where $\mathcal{R}'$ differs from $\mathcal{R}$ only by the sign flip of a row or column.

A particularly simple example of such a flow occurs for a single Dirac fermion. In Appendix B, we show explicitly how the absorption of a boundary Majorana mode exchanges the boundary conditions (1.3) and (1.4).

---

[4]This normalisation for the axial boundary state was recently advocated in [48] to ensure compatibility with the vector-like boundary conditions, although the connection to the mod 2 anomaly was not made.

## 4.2 Non-Primitive Boundary States

We now turn to the second subtlety in the RG flows. We have seen that turning on a single, relevant operator in the UV breaks $U(1)^N \to U(1)^{N-1}$. However, this is not the full story. There is also a remnant discrete symmetry, so that

$$U(1)^N \to U(1)^{N-1} \times \mathbb{Z}_n$$

Here, the integer $n$ is the same one introduced in (4.6), which measures the failure of $\rho$ to be a primitive vector.

This discrete symmetry $\mathbb{Z}_n$ is preserved along the RG flow. However, one finds that the naïve IR boundary state is *not* invariant under the full $\mathbb{Z}_n$ symmetry. To rectify this, the infra-red boundary state must be a linear sum of states of the form (2.8) such that the overall sum is $\mathbb{Z}_n$ invariant. The different states in this sum have the same $\mathcal{R}_{IR}$ charge matrix, but differ in their theta angles. This then shows up in the infra-red central charge, with each state in the sum contributing a factor of $\sqrt{\text{Vol}(\Lambda[\mathcal{R}_{IR}])}$. We'll discuss this further in Section 4.3.

To put some flesh on these ideas, we will need to understand how the $\mathbb{Z}_n$ symmetry acts on our candidate infra-red boundary state (2.8),

$$|\theta; \mathcal{R}_{IR}\rangle = g_{\mathcal{R}} \sum_{\lambda \in \Lambda[\mathcal{R}_{IR}]} e^{i\gamma(\lambda)} e^{i\theta \cdot \lambda} \| \lambda, -\mathcal{R}_{IR}\lambda \rangle\!\rangle \tag{4.8}$$

Under a transformation by $k \in \mathbb{Z}_n$, the sole effect on the infra-red boundary state is is to shift the theta angles $\theta_i$ by

$$\frac{\theta}{2\pi} \mapsto \frac{\theta}{2\pi} + \frac{2k}{\rho^2}\rho$$

The unbroken subgroup of $\mathbb{Z}_n$ will consist of those $k$ for which this shift has no effect on the boundary state. To determine when this is the case, we note that the theta angles in (4.8) appear in the phase $e^{i\theta \cdot \lambda}$, which means that $\theta/2\pi$ is naturally defined mod $\Lambda[\mathcal{R}_{IR}]^\star$. Therefore, the above shift is trivial whenever $(2k/\rho^2)\rho \in \Lambda[\mathcal{R}_{IR}]^\star$. We introduce the integer $m \geq 1$, defined as the least integer such that

$$\frac{2m}{\rho^2}\rho \in \Lambda[\mathcal{R}_{IR}]^\star \tag{4.9}$$

Then $m$ divides $n$, and in the infra-red, the $\mathbb{Z}_n$ symmetry is spontaneously broken by the boundary state (4.8) to

$$\mathbb{Z}_n \to \mathbb{Z}_{n/m} \tag{4.10}$$

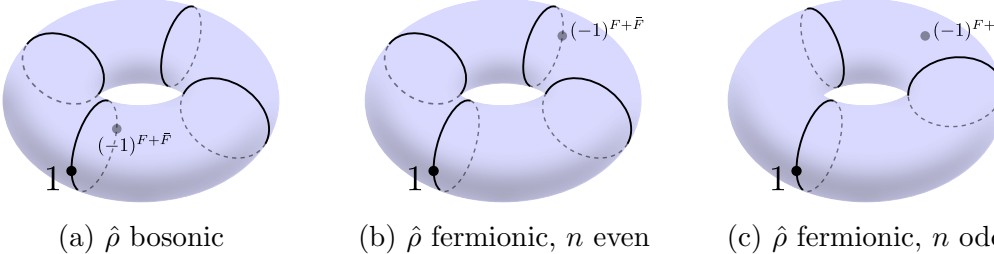

(a) $\hat\rho$ bosonic      (b) $\hat\rho$ fermionic, $n$ even      (c) $\hat\rho$ fermionic, $n$ odd

**Figure 3**: How $(-1)^{F+\bar F}$ sits in $U(1)^{N-1} \times \mathbb{Z}_n \subset U(1)^N$.

Just like the criterion for whether a boundary Majorana mode appears, the integer $m$ can also be determined in terms of basic properties of $\rho$. It is given by

$$m = \begin{cases} n & \text{if } \hat\rho \text{ is bosonic} \\ n/\gcd(n,2) & \text{if } \hat\rho \text{ is fermionic} \end{cases} \tag{4.11}$$

The upshot is that there are only two possibilities for the residual discrete symmetry:

$$\mathbb{Z}_n \to 1 \text{ or } \mathbb{Z}_2$$

As we now explain, the presence or absence of the unbroken $\mathbb{Z}_2$ has a simple physical explanation: it remains unbroken when fermion parity $(-1)^{F+\bar F}$ forces it to. This is illustrated in Figure 3. Here we have depicted the UV $U(1)^N$ symmetry group, the $U(1)^{N-1} \times \mathbb{Z}_n$ subgroup left unbroken by the perturbation, and the location of fermion parity in relation to both. From Section 3.2, we know that $(-1)^{F+\bar F}$ always lies within $U(1)^N$. But by definition, it only lies in $U(1)^{N-1} \times \mathbb{Z}_n$ if $\rho$ is bosonic. This information alone is enough to fix the location of $(-1)^{F+\bar F}$ – it belongs to the coset $k = 0$ in case (a), to $k = n/2$ in case (b), and to none of them in case (c).

The transformation $(-1)^{F+\bar F}$ is a sacrosanct symmetry. Being part of the conformal group, it is automatically preserved by all the boundary states (2.8). This means that if ever a coset $k$ contains $(-1)^{F+\bar F}$, that coset is automatically preserved. We see that this happens precisely in case (b), which coincides with condition (4.11) for a $\mathbb{Z}_2$ to remain unbroken. In other words, the discrete $\mathbb{Z}_n$ is completely broken, except for the part fermion parity forces to stay unbroken.

Finally, we should ask: what is the infra-red boundary state? Clearly the boundary state must be invariant under the $\mathbb{Z}_n$ symmetry. The obvious choice is to take a non-fundamental boundary state, consisting of a sum over the various theta angles

$$|\text{IR}\rangle = \sum_{k=0}^{m-1} |\theta + \tfrac{2k}{\rho^2}\rho; \mathcal{R}_{IR}\rangle \tag{4.12}$$

This captures the symmetry breaking (4.10) in a minimal way, with the least possible number of fundamental boundary states in the sum. The boundary central charge picks up a contribution from each term in (4.12). Furthermore, it turns out that, in some examples, any attempt to add further boundary states to this sum results in a violation of the $g$-theorem. This gives credence to this minimalist conjecture. The result is that, if there is no emergent Majorana zero mode, then the infra-red central charge is given by

$$g_{IR} = m \operatorname{Vol}(\Lambda[\mathcal{R}_{IR}]) \tag{4.13}$$

If, in addition, there is an emergent Majorana mode then we have an additional factor of $\sqrt{2}$, as in (4.5).

**An Example**

The simplest example of a non-primitive boundary state can be found in the two fermion theory with $g_{\mathcal{R}} = \sqrt{5}$.

A glance at the table in Section 4.1 shows that there are two operators with dimension $L_0 = \frac{1}{5}$, characterised by the primitive vectors

$$\hat{\rho}_1 = \left(\frac{1}{5}, \frac{3}{5}\right) \quad \text{and} \quad \hat{\rho}_2 = \left(\frac{3}{5}, -\frac{1}{5}\right)$$

Deforming by either of these operators breaks $U(1)^2 \to U(1)$.

There are also two operators with dimension $L_0 = \frac{4}{5}$, which have $\rho_a = 2\hat{\rho}_a$, with $a = 1, 2$. Deforming by either of these operators breaks $U(1)^2 \to U(1) \times \mathbb{Z}_2$.

From the previous table, we see that deforming by either $\hat{\rho}_a$ or $\rho_a = 2\hat{\rho}_a$ results in the same infra-red charge matrix $\mathcal{R}_{IR}$. This is a trivial, non-chiral state with $\operatorname{Vol}(\mathcal{R}_{IR}) = 1$. However, when we deform by the non-primitive vector, we must sum over two infra-red boundary states to preserve the $\mathbb{Z}_2$. The net result is that the two deformations give different infra-red central charges

$$\begin{aligned} \hat{\rho}_a &\Rightarrow & g_{IR} &= 1 \\ \rho_a = 2\hat{\rho}_a &\Rightarrow & g_{IR} &= 2 \end{aligned}$$

## 4.3 The Boundary Central Charge

All the ingredients are now in place to determine the boundary state in the infra-red and its central charge. We start from a UV boundary state $|\theta; \mathcal{R}_{UV}\rangle$, with

$$g_{UV} = \sqrt{\operatorname{Vol}(\Lambda_{UV})}$$

where $\Lambda_{UV} = \Lambda[\mathcal{R}_{UV}]$. We deform by a relevant, bosonic, boundary operator characterised by $\rho \in \Lambda_{UV}^{\star}$. The IR boundary state is then determined by several factors:

- The infra-red charge matrix $\mathcal{R}_{IR}$, given by (4.2). It contributes a factor of $\sqrt{\mathrm{Vol}(\Lambda_{IR})}$ to the central charge, where $\Lambda_{IR} = \Lambda[\mathcal{R}_{IR}]$.

- If the boundary state changes SPT class, as determined by (4.7), there is an emergent Majorana mode on the boundary. This increases the infra-red central charge by $\sqrt{2}$.

- If $\rho = n\hat{\rho}$ is not primitive, there is naïvely a discrete symmetry breaking pattern in which $\mathbb{Z}_n \to \mathbb{Z}_{n/m}$ with $m$ determined by (4.11). To avoid spontaneous breaking of this symmetry, we must sum over $m$ boundary states. This increases the central charge by $m$.

To compute the IR central charge, we need the relation between the volumes of the IR and UV charge lattices. This will be computed in Section 5: it turns out to be

$$\mathrm{Vol}(\Lambda_{IR}) = \hat{\rho}^2 \, \mathrm{Vol}(\Lambda_{UV}) \times \begin{cases} \frac{1}{2} & \text{if } \hat{\rho} \text{ is bosonic} \\ 1 & \text{if } \hat{\rho} \text{ is fermionic} \end{cases} \tag{4.14}$$

We can now consider the following three types of flows.

- Bosonic flows that preserve the SPT class

  Flows that leave the SPT class unchanged are initiated by operators with charge $\rho = n\hat{\rho}$ with $\hat{\rho}$ bosonic, and $n$ any integer. The discrete symmetry breaking pattern is $\mathbb{Z}_n \to 1$, and the boundary state takes the form

$$|\theta; \mathcal{R}_{UV}\rangle \; \to \; \sum_{k=0}^{n-1} |\theta + \tfrac{2k}{\rho^2}\rho; \mathcal{R}_{IR}\rangle$$

  In this case, the ratio of IR to UV central charges is given by

$$\frac{g_{IR}}{g_{UV}} = n \sqrt{\frac{\mathrm{Vol}(\Lambda_{IR})}{\mathrm{Vol}(\Lambda_{UV})}} = \sqrt{\rho^2/2}$$

- Bosonic flows that change the class

  Flows that flip the SPT class are initiated by operators with charge $\rho = n\hat{\rho}$ with $\hat{\rho}$ fermionic. If this operator is bosonic then $n$ is even. This time the discrete symmetry breaking is $\mathbb{Z}_n \to \mathbb{Z}_2$, and

$$|\theta; \mathcal{R}_{UV}\rangle \; \to \; \sqrt{2} \sum_{k=0}^{\frac{n}{2}-1} |\theta + \tfrac{2k}{\rho^2}\rho; \mathcal{R}_{IR}\rangle$$

The ratio of IR to UV central charge is now

$$\frac{g_{IR}}{g_{UV}} = \sqrt{2} \times \frac{n}{2} \sqrt{\frac{\mathrm{Vol}(\Lambda_{IR})}{\mathrm{Vol}(\Lambda_{UV})}} = \sqrt{\rho^2/2}$$

- Fermionic flows

  Finally, if we start in the UV with an extra Majorana mode then we can perturb by a fermionic operator with charge $\rho = n\hat{\rho}$ with $\hat{\rho}$ fermionic and $n$ odd. The discrete symmetry breaking is $\mathbb{Z}_n \to 1$. We also know that the flow flips the SPT class, since $\hat{\rho}$ is fermionic. The flow of boundary states is now

  $$\sqrt{2}\,|\theta; \mathcal{R}_{UV}\rangle \;\to\; \sum_{k=0}^{n-1} |\theta + \tfrac{2k}{\rho^2}\rho; \mathcal{R}_{IR}\rangle$$

  and the ratio of IR to UV central charges is

  $$\frac{g_{IR}}{g_{UV}} = \frac{1}{\sqrt{2}} \times n \sqrt{\frac{\mathrm{Vol}(\Lambda_{IR})}{\mathrm{Vol}(\Lambda_{UV})}} = \sqrt{\rho^2/2}$$

**The central charge relation**

Importantly, we find the same ratio of central charges for each of the three types of RG flows described above. Moreover, we recognise $L_0 = \rho^2/2$ as the dimension of the UV operator $\mathcal{O}$ that initiates the RG flow. We learn that

$$g_{IR} = g_{UV}\,\sqrt{\dim \mathcal{O}}$$

This is the formula (1.5) advertised in the introduction. Since the UV operator is necessarily relevant, we have $\rho^2 < 2$. This ensures that $g_{IR} < g_{UV}$, and the $g$-theorem is obeyed.

**More General RG Flows**

In our discussion above, we have restricted attention to RG flows initiated by operators with a definite charge under $U(1)^N$. This ensures that the original symmetry is broken to $U(1)^{N-1}$, which allowed us to identify the infra-red state (4.2).

More generally, we could deform by turning on superpositions of such operators with different $\rho$. The resulting RG flows can be understood by following first one deformation, then the other. For certain UV boundary states, we can reach IR states this way which cannot be reached by turning on one operator alone.

An example is provided by the $g = 9$ four-fermion state

$$\mathcal{R}_{UV} = \begin{pmatrix} 0 & -\frac{2}{3} & \frac{1}{3} & \frac{2}{3} \\ -\frac{2}{3} & 0 & -\frac{2}{3} & \frac{1}{3} \\ -\frac{1}{3} & -\frac{2}{3} & 0 & -\frac{2}{3} \\ \frac{2}{3} & -\frac{1}{3} & -\frac{2}{3} & 0 \end{pmatrix}$$

Deformations by charge eigenstates will take us to IR states with $g = 9, 6, 3$. However, they will not take us the trivial state with $g = 1$. This can be reached by a more general perturbation, such as by chaining together the flows $9 \to 3 \to 1$.

## 5 RG Flows: Proofs

In Section 4 we stated a number of results without proof. Here we give the proofs.

### 5.1 The UV Symmetry

Given the charge matrix $\mathcal{R}_{UV}$, the $U(1)^N$ symmetry group preserved in the UV is

$$U(1)^N = \left\{ (e^{2\pi i t_\alpha Q_{\alpha i}}, e^{2\pi i t_\alpha \bar{Q}_{\alpha i}}) : t \in \mathbb{R}^N \right\}$$

where $Q_{\alpha i}$ and $\bar{Q}_{\alpha i}$ are the UV charge assignments. Using the definition $\mathcal{R}_{UV} = \bar{Q}^{-1}Q$, this group can be parametrised in the more useful form

$$U(1)^N = \left\{ (e^{2\pi i x}, e^{2\pi i \mathcal{R}_{UV} x}) : x \in \mathbb{R}^N \right\}$$

The symmetry parameter $x$ is naturally valued in $\mathbb{R}^N / \Lambda_{UV}$.

Given the boundary operator charge $\rho \in \Lambda^\star_{UV}$, we first wish to determine how much of $U(1)^N$ remains unbroken by the perturbation. Under the $U(1)^N$ transformation with parameter $x$, the boundary operator picks up a phase of $e^{2\pi i x \cdot \rho}$. This means that perturbing operator is invariant when

$$x \cdot \rho \in \mathbb{Z}$$

Let us write $\rho = n\hat{\rho}$ with $n \geq 1$ and $\hat{\rho}$ primitive in $\Lambda^\star_{UV}$. Because $\hat{\rho}$ is primitive, we can introduce a special basis for $\Lambda_{UV}$ with

$$\Lambda_{UV} = \text{span}\{\lambda_1, \ldots, \lambda_N\}$$
$$\lambda_1 \cdot \hat{\rho} = 1$$
$$\{\lambda_2, \ldots, \lambda_N\} \cdot \hat{\rho} = 0$$

Writing $x$ in components with respect to this basis, the above condition for invariance becomes

$$x_1 \in \tfrac{1}{n}\mathbb{Z} \qquad x_2, \ldots, x_N \in \mathbb{R}$$

Since the variables $x_i$ are defined mod 1, we see that the first variable $x_1$ parametrises a discrete $\mathbb{Z}_n$, while the remaining variables $x_2, \ldots x_N$ parametrise a $U(1)^{N-1}$. In other words, the $U(1)^N$ is broken to $U(1)^{N-1} \times \mathbb{Z}_n$, with the coset corresponding to $k \in \mathbb{Z}_n$ being all those transformations with parameter $x$ obeying

$$x \cdot \rho = k$$

This puts us in a position to justify the form of the IR charge matrix. The $U(1)^{N-1}$ corresponds to those transformations with

$$x \in \rho^\perp$$

The statement that these are also preserved by the IR boundary state is that

$$\mathcal{R}_{IR} x = \mathcal{R}_{UV} x$$

This immediately leads to (4.1).

## 5.2   The Infra-red Lattice

Given that the IR charge matrix takes the form

$$\mathcal{R}_{IR} = \mathcal{R}_{UV} \mathrm{Ref}_\rho$$

where $\mathrm{Ref}_\rho$ denotes reflection along $\rho$, it follows immediately that both $\Lambda_{IR}$ and $\Lambda_{UV}$ share the same intersection with $\rho^\perp$, the hyperplane perpendicular to $\rho$:

$$\Lambda_{IR} \cap \rho^\perp = \Lambda_{UV} \cap \rho^\perp = \mathrm{span}\,\{\lambda_2, \ldots, \lambda_N\}$$

It follows that there is a basis for $\Lambda_{IR}$ consisting of

$$\Lambda_{IR} = \mathrm{span}\left\{\tilde{\lambda}_1, \lambda_2, \ldots, \lambda_N\right\}$$

Here $\tilde{\lambda}_1$ is the single, remaining basis vector of $\Lambda_{IR}$, which remains to be determined. In fact, all we shall need to know about it is provided by the following claim:

**Claim:** The extra basis vector $\tilde{\lambda}_1$ of $\Lambda_{IR}$ is of the form

$$\tilde{\lambda}_1 = \left\{ \begin{array}{ll} \tfrac{1}{2} & \text{if } \hat{\rho} \text{ is bosonic} \\ 1 & \text{if } \hat{\rho} \text{ is fermionic} \end{array} \right\} \hat{\rho} \quad \mod \rho^\perp$$

**Proof:** A general vector $\lambda \in \mathbb{R}^N$ can be written in the form

$$\lambda = a\hat{\rho} + \eta \quad \text{with } a \in \mathbb{R} \text{ and } \eta \in \rho^\perp$$

We wish to determine the constraints on $a$ and $\eta$ that arise from insisting $\lambda \in \Lambda_{IR}$. In particular, we are particularly interested in the quantisation condition on $a$. The first constraint is that $\lambda$ must be an integer vector, which we call $x$:

$$a\hat{\rho} + \eta = x \tag{5.1}$$

The second constraint is that $(\mathcal{R}_{UV}\text{Ref}_\rho)\lambda$ must be an integer vector, which we call $y$. Using the fact that $\text{Ref}_\rho$ flips $\hat{\rho}$ while leaving $\eta$ unaffected, we have

$$\mathcal{R}_{UV}(a\hat{\rho} - \eta) = y \quad \Rightarrow \quad a\hat{\rho} - \eta = \mathcal{R}_{UV}^{-1}y \tag{5.2}$$

To proceed, we take the sum and difference of (5.1) and (5.2). First, the sum tells us that

$$2a\hat{\rho} = x + \mathcal{R}_{UV}^{-1}y \quad \text{with } x, y \in \mathbb{Z}^N \tag{5.3}$$

We take the inner product with the basis vector $\lambda_1 \in \Lambda_{UV}$, which obeys $\lambda_1 \cdot \hat{\rho} = 1$. On the right-hand side, we have $\lambda_1 \cdot x \in \mathbb{Z}$ since both $\lambda_1$ and $x$ are integral. Furthermore, $\lambda_1 \cdot \mathcal{R}_{UV}^{-1}y \in \mathbb{Z}$ since this is equal to $\mathcal{R}_{UV}\lambda_1 \cdot y$ and $\mathcal{R}_{UV}\lambda_1$ is integral by definition of $\Lambda_{UV}$. We learn that

$$2a \in \mathbb{Z}$$

Next we invoke the fact that $\hat{\rho}$ lies in $\Lambda_{UV}^\star = \mathbb{Z}^N + \mathcal{R}_{UV}^{-1}\mathbb{Z}^N$. This means that $\hat{\rho}$ can be written in the form $\rho = v + \mathcal{R}_{UV}^{-1}w$ for two further integer vectors $v$ and $w$. The equation (5.3) then becomes

$$2a(v + \mathcal{R}_{UV}^{-1}w) = x + \mathcal{R}_{UV}^{-1}y \tag{5.4}$$

It is obvious that one solution to this equation for $(x, y)$ is $x = 2av$ and $y = 2aw$. However, this is not the unique solution since we still have the freedom to shift by any integer solution to $x + \mathcal{R}_{UV}^{-1}y = 0$. These are precisely $(x, y) = (\zeta, -\mathcal{R}_{UV}\zeta)$ for $\zeta \in \Lambda_{UV}$. The general solution to (5.4) is then

$$x = 2av + \zeta \quad \text{and} \quad y = 2aw - \mathcal{R}_{UV}\zeta$$

The above equations were derived by taking the sum of (5.1) and (5.2). Next we take the difference. This gives

$$2\eta = x - \mathcal{R}_{UV}^{-1}y \quad \Rightarrow \quad \eta = a(v - \mathcal{R}_{UV}^{-1}w) + \zeta$$

The variables $a \in \frac{1}{2}\mathbb{Z}$ and $\zeta \in \Lambda_{UV}$ are further constrained by the requirement that $\eta \in \rho^\perp$. Taking the inner product with $\hat{\rho}$ and setting this to zero gives

$$\zeta \cdot \hat{\rho} = -a\left[(v - \mathcal{R}_{UV}^{-1}) \cdot \hat{\rho}\right] = -a(v^2 - w^2)$$

The left-hand side is an integer. But $a$ can be either integer of half-integer. Clearly the half-integer values can only occur when $v^2 - w^2$ is even which, in turn, requires $\sum_{i=1}^{N}(v_i + w_i)$ to be even. But this is precisely the fermionic parity of $\hat{\rho}$.

To see this, note that $\Lambda[\mathcal{R}]^\star = \mathbb{Z}^N + \mathcal{R}^{-1}\mathbb{Z}^N$ has a simple physical interpretation: all boundary operators can be made by taking suitably regularised products of holomorphic and antiholomorphic fermion fields as they approach the boundary. A product of $n_i$ copies of $\psi_i(z)$ and $m_i$ copies of $\bar{\psi}_i(z)$ would give rise to a boundary operator with charge $\rho = n + \mathcal{R}^{-1}m$. It's clear that the fermion parity of this operator is

$$(-1)^{n_1 + \cdots + n_N + m_1 + \cdots + m_N} \tag{5.5}$$

Using the properties of the fermion vector $f$, defined in (3.7), this can easily be shown to agree with the earlier characterisation $(-1)^{f \cdot \rho}$. Using the fact that $\hat{\rho} = v + \mathcal{R}_{UV}^{-1}w$, we then learn that

$$a \in \begin{cases} \frac{1}{2}\mathbb{Z} & \text{if } \hat{\rho} \text{ is bosonic} \\ \mathbb{Z} & \text{if } \hat{\rho} \text{ is fermionic} \end{cases}$$

The conditions derived above are necessary for $\lambda = a\hat{\rho} + \eta$ to lie in $\Lambda_{IR}$. The same derivation can also be followed backwards to show they are sufficient. All of which means that we finally have an expression for our last remaining basis vector of $\Lambda_{IR}$;

$$\tilde{\lambda}_1 = \left\{ \begin{array}{ll} \frac{1}{2} & \text{if } \hat{\rho} \text{ is bosonic} \\ 1 & \text{if } \hat{\rho} \text{ is fermionic} \end{array} \right\} \hat{\rho} + \eta \tag{5.6}$$

for some $\eta \in \rho^\perp$ whose value is unimportant. This completes the proof of the claim.

We are now in a position to compute the volume of $\Lambda_{IR}$. This is

$$\text{Vol}(\Lambda_{IR}) = \text{Vol}(\tilde{\lambda}_1, \lambda_2, \ldots, \lambda_N) = \text{Vol}((\hat{\rho} \cdot \tilde{\lambda}_1)\lambda_1, \lambda_2, \ldots, \lambda_N)$$

We therefore find

$$\text{Vol}(\Lambda_{IR}) = \hat{\rho} \cdot \tilde{\lambda}_1 \, \text{Vol}(\Lambda_{UV}) = \left\{ \begin{array}{ll} \frac{1}{2} & \text{if } \hat{\rho} \text{ is bosonic} \\ 1 & \text{if } \hat{\rho} \text{ is fermionic} \end{array} \right\} \hat{\rho}^2 \, \text{Vol}(\Lambda_{UV}) \tag{5.7}$$

This provides the justification for (4.14).

The above result also allows us to determine the integer $m$ which governs the amount of discrete symmetry breaking. Under a general $U(1)^N$ transformation with parameter $x$, we have

$$|\theta; \mathcal{R}_{IR}\rangle \mapsto g_{\mathcal{R}} \sum_{\lambda \in \Lambda[\mathcal{R}_{IR}]} e^{i\gamma(\lambda)} e^{\theta \cdot \lambda} e^{2\pi i x \cdot \lambda} e^{2\pi i (\mathcal{R}_{UV} x) \cdot (-\mathcal{R}_{IR} \lambda)} \| \lambda, -\mathcal{R}_{IR} \lambda \rangle\!\rangle$$

We see that the effect of this is to shift the theta angles $\theta_i$ of the infra-red boundary state by

$$\frac{\theta}{2\pi} \mapsto \frac{\theta}{2\pi} + \left(\mathbb{1} - \mathcal{R}_{UV}^{-1} \mathcal{R}_{IR}\right) x = \frac{\theta}{2\pi} + \frac{2(x \cdot \rho)}{\rho^2} \rho$$

where, in the second equality, we have used the expression (4.2) for $\mathcal{R}_{IR}$. We see explicitly that the theta angles are invariant under the preserved $U(1)^{N-1}$ symmetry defined by those $x$ with $x \cdot \rho = 0$. But what of the discrete $\mathbb{Z}_n$ symmetry? A transformation by $k \in \mathbb{Z}_n$ is enacted by any $x$ for which $x \cdot \rho = k$, and shifts the theta angles by

$$\frac{\theta}{2\pi} \mapsto \frac{\theta}{2\pi} + \frac{2k}{\rho^2} \rho$$

The theta angles in (4.8) appear in the phase $e^{i\theta \cdot \lambda}$, which means that they are naturally valued mod $2\pi \Lambda^{\star}[\mathcal{R}_{IR}]$. Therefore the transformation above leaves the theta angles invariant whenever

$$\frac{2k}{\rho^2} \rho \in \Lambda^{\star}[\mathcal{R}_{IR}]$$

The above condition will be satisfied if the LHS gives an integer when dotted with every basis vector of $\Lambda_{IR}$. Of these, the last $N-1$ vectors $\lambda_2, \ldots, \lambda_N$ give zero. Thus a constraint only arises by dotting with $\tilde{\lambda}_1$. Recalling the definition (5.6) of $\tilde{\lambda}_1$, this gives

$$\left\{ \begin{array}{ll} \frac{1}{2} & \text{if } \hat{\rho} \text{ is bosonic} \\ 1 & \text{if } \hat{\rho} \text{ is fermionic} \end{array} \right\} \cdot \frac{1}{n} \cdot 2k \in \mathbb{Z}$$

It is now straightforward to read off the quantisation condition on $k$. It must be a multiple of $m$, where $m$ is defined by

$$m = \left\{ \begin{array}{ll} n & \text{if } \hat{\rho} \text{ is bosonic} \\ n/\gcd(n,2) & \text{if } \hat{\rho} \text{ is fermionic} \end{array} \right.$$

This is the statement of (4.11).

## 5.3 The Emergent Majorana Mode

The final missing ingredient is to determine when a boundary Majorana mode arises. As explained in section 4, this happens when the UV and IR charge matrices lie in different classes, which is detected by the ground degeneracy of bulk states (4.4),

$$G[\mathcal{R}_{UV}, \mathcal{R}_{IR}] = \frac{\sqrt{\mathrm{Vol}(\Lambda_{UV})\,\mathrm{Vol}(\Lambda_{IR})}}{\mathrm{Vol}(\Lambda[\mathcal{R}_{UV}, \mathcal{R}_{IR}])}\sqrt{2}$$

where the factor of $\sqrt{2}$ comes from the truncated determinant in (4.4), using the expression (4.2) for $\mathcal{R}_{IR}$. Clearly, we need to compute the volume of the intersection lattice

$$\Lambda[\mathcal{R}_{UV}, \mathcal{R}_{IR}] = \left\{\lambda \in \mathbb{Z}^N : \mathcal{R}_{UV}\lambda = \mathcal{R}_{IR}\lambda \in \mathbb{Z}^N\right\}$$

First, we can write

$$\Lambda[\mathcal{R}_{UV}, \mathcal{R}_{IR}] = \left\{\lambda \in \mathbb{Z}^N : \lambda \cdot \rho = 0 \text{ and } \mathcal{R}_{UV}\lambda \in \mathbb{Z}^N\right\} = \Lambda_{UV} \cap \rho^\perp$$

But using the basis of $\Lambda_{UV}$, the intersection lattice takes the particularly simple form

$$\Lambda[\mathcal{R}_{UV}, \mathcal{R}_{IR}] = \mathrm{span}\left\{\lambda_2, \ldots, \lambda_N\right\}$$

To determine the volume of this intersection lattice, we need to take the above basis and add a unit vector orthogonal to them all. This vector is $\hat{\rho}/\sqrt{\hat{\rho}^2}$, so

$$\mathrm{Vol}(\Lambda[\mathcal{R}_{UV}, \mathcal{R}_{IR}]) = \mathrm{Vol}\left(\hat{\rho}/\sqrt{\hat{\rho}^2}, \lambda_2, \ldots, \lambda_N\right)$$

But we could equally well shift the first basis vector by any element in $\rho^\perp$. Using the property $\lambda_1 \cdot \hat{\rho} = 1$, we then have

$$\mathrm{Vol}(\Lambda[\mathcal{R}_{UV}, \mathcal{R}_{IR}]) = \mathrm{Vol}\left(\sqrt{\hat{\rho}^2}\lambda_1, \lambda_2, \ldots, \lambda_N\right) = \sqrt{\hat{\rho}^2}\,\mathrm{Vol}(\Lambda_{UV})$$

If we now put this together with our expression (5.7) for the volume of $\Lambda_{IR}$, we have the simple result

$$G[\mathcal{R}_{UV}, \mathcal{R}_{IR}] = \begin{cases} 1 & \text{if } \hat{\rho} \text{ is bosonic} \\ \sqrt{2} & \text{if } \hat{\rho} \text{ is fermionic} \end{cases}$$

which establishes (4.7).

# A  A Higher Pythagorean Triple

For $N = 2$ Dirac fermions, the chiral boundary conditions are in one-to-one correspondence with Pythagorean triples [35]. With the Euclid parameterisation (2.10) with $p = 4$ and $q = 1$, we have the Pythagorean triple $8^2 + 15^2 = 17^2$. The charge matrix is

$$\mathcal{R}_{UV} = \frac{1}{17} \begin{pmatrix} 8 & 15 \\ -15 & 8 \end{pmatrix}$$

This boundary state has $g_{UV}^2 = 17$. Various RG flows initiated by bosonic operators are summarised in the following table:

| $\rho$ | $L_0$ | $\mathcal{R}_{IR}$ | Majorana? | $g_{IR}^2$ |
|---|---|---|---|---|
| $\left(\frac{5}{17}, \frac{3}{17}\right)$ | $\frac{1}{17}$ | $\begin{pmatrix} -1 & 0 \\ 0 & 1 \end{pmatrix}$ | No | 1 |
| $\left(\frac{3}{17}, -\frac{5}{17}\right)$ | $\frac{1}{17}$ | $\begin{pmatrix} 1 & 0 \\ 0 & -1 \end{pmatrix}$ | No | 1 |
| $\left(\frac{8}{17}, -\frac{2}{17}\right)$ | $\frac{2}{17}$ | $\begin{pmatrix} 0 & 1 \\ 1 & 0 \end{pmatrix}$ | Yes | 2 |
| $\left(\frac{2}{17}, \frac{8}{17}\right)$ | $\frac{2}{17}$ | $\begin{pmatrix} 0 & -1 \\ -1 & 0 \end{pmatrix}$ | Yes | 2 |
| $\left(\frac{13}{17}, \frac{1}{17}\right)$ | $\frac{5}{17}$ | $\frac{1}{5}\begin{pmatrix} -3 & 4 \\ 4 & 3 \end{pmatrix}$ | No | 5 |
| $\left(\frac{11}{17}, -\frac{7}{17}\right)$ | $\frac{5}{17}$ | $\frac{1}{5}\begin{pmatrix} 3 & 4 \\ 4 & -3 \end{pmatrix}$ | No | 5 |
| $\left(\frac{7}{17}, \frac{11}{17}\right)$ | $\frac{5}{17}$ | $\frac{1}{5}\begin{pmatrix} -3 & -4 \\ -4 & 3 \end{pmatrix}$ | No | 5 |
| $\left(\frac{1}{17}, -\frac{13}{17}\right)$ | $\frac{5}{17}$ | $\frac{1}{5}\begin{pmatrix} 3 & -4 \\ -4 & -3 \end{pmatrix}$ | No | 5 |
| $\left(\frac{18}{17}, \frac{4}{17}\right)$ | $\frac{10}{17}$ | $\frac{1}{5}\begin{pmatrix} -4 & 3 \\ 3 & 4 \end{pmatrix}$ | Yes | 10 |
| $\left(\frac{14}{17}, -\frac{12}{17}\right)$ | $\frac{10}{17}$ | $\frac{1}{5}\begin{pmatrix} 4 & 3 \\ 3 & -4 \end{pmatrix}$ | Yes | 10 |
| $\left(\frac{12}{17}, \frac{14}{17}\right)$ | $\frac{10}{17}$ | $\frac{1}{5}\begin{pmatrix} -4 & -3 \\ -3 & 4 \end{pmatrix}$ | Yes | 10 |
| $\left(\frac{4}{17}, -\frac{18}{17}\right)$ | $\frac{10}{17}$ | $\frac{1}{5}\begin{pmatrix} 4 & -3 \\ -3 & -4 \end{pmatrix}$ | Yes | 10 |
| $\left(\frac{21}{17}, -\frac{1}{17}\right)$ | $\frac{13}{17}$ | $\frac{1}{13}\begin{pmatrix} -5 & 12 \\ 12 & 5 \end{pmatrix}$ | No | 13 |
| $\left(\frac{19}{17}, -\frac{9}{17}\right)$ | $\frac{13}{17}$ | $\frac{1}{13}\begin{pmatrix} 5 & 12 \\ 12 & -5 \end{pmatrix}$ | No | 13 |
| $\left(\frac{19}{17}, -\frac{9}{17}\right)$ | $\frac{13}{17}$ | $\frac{1}{13}\begin{pmatrix} -5 & -12 \\ -12 & 5 \end{pmatrix}$ | No | 13 |
| $\left(\frac{1}{17}, \frac{21}{17}\right)$ | $\frac{13}{17}$ | $\frac{1}{13}\begin{pmatrix} 5 & -12 \\ -12 & -5 \end{pmatrix}$ | No | 13 |

This table lists relevant, bosonic operators and their end points under RG. For simplicity, we restrict to primitive $\rho$, so that there are no discrete symmetries and the infra-red central charge $g_{IR}$ is determined solely by $\mathcal{R}_{IR}$ and the existence of a boundary Majorana fermion.

Note that the dimensions of the relevant operators take the form

$$L_0 = \frac{m^2 + n^2}{p^2 + q^2} \qquad p, q, m, n \in \mathbb{Z}$$

where, for us, $p = 4$ and $q = 1$. Turning on an operator with this dimension takes us to a new state with primitive charges $m, n$ in (2.10). This same property holds for all boundary states with $N = 2$ fermions. We do not know of such a simple pattern for $N \geq 4$.

## B The Boundary Majorana Mode

In this appendix, we explain how a boundary Majorana mode interacts with the bulk fermions. Very similar calculations can be found in [3, 18] and related analysis in [45, 46].

### A Fermion on a Half Line

We start with a single Majorana fermion $\xi$ on a half-line, interacting with a quantum mechanical Majorana fermion $\chi$ sitting on the boundary. It is simplest if we unfold the system, leaving us with a single right-moving Majorana-Weyl fermion on a line, interacting with a Majorana impurity at the origin. The Hamiltonian is

$$\mathcal{H} = \frac{i}{2}\chi\partial_t\chi + \int dx \left[ \frac{i}{2}\xi\partial_+\xi + i\sqrt{2m}\delta(x)\xi(x)\chi \right]$$

The coupling between bulk and boundary is simply a quadratic term, set by a mass scale $m$. As we will see, only modes with momentum $k \ll m$ are significantly affected by the impurity.

To proceed, it is useful to temporarily smooth out the delta-function coupling. We replace the Hamiltonian with

$$\mathcal{H} = \frac{i}{2}\chi\partial_t\chi + \int dx \left[ \frac{i}{2}\xi\partial_+\xi + i\sqrt{2m}f(x)\xi\chi \right]$$

where $f(x)$ is some function localised around the origin, with support in $x \in [-\epsilon, +\epsilon]$, and with $\int dx \, f(x) = 1$. The equations of motion are:

$$\partial_t \chi = \sqrt{2m} \int dx \, f\xi$$
$$\partial_+ \xi = -\sqrt{2m} f\chi$$

Modes with energy $k$ have time dependence $e^{-ikt}$. (All fermions are subject to a reality condition, but the equations of motion are linear so we can work with complex objects and take the real part at the end.) The equations of motion become

$$-ik\chi = \sqrt{2m} \int dx \, f\xi$$
$$-ik\xi + \partial_x \xi = -\sqrt{2m} f\chi$$

We are interested in modes with $k \ll 1/\epsilon$, which ensures that they don't probe the microscopic details of the function $f(x)$. Near the origin, $|x| \leq \epsilon$, the second equation can then be replaced by $\partial_x \xi = -\sqrt{2m} f\chi$. We integrate the second equation in the asymptotic regions, and join them up to find

$$\xi(x) = \begin{cases} e^{ikx} & x < -\epsilon \\ 1 - \sqrt{2m} F(x)\chi & \text{otherwise} \\ (1 - \sqrt{2m}\chi)e^{ikx} & x > \epsilon \end{cases} \tag{B.1}$$

where $F(x)$ is a step function that goes smoothly from 0 to 1, with $F'(x) = f(x)$. Substituting this into the equation for $\chi$ gives us a consistency condition,

$$-ik\chi = \sqrt{2m}\left(1 - \sqrt{m/2}\,\chi\right)$$

which has the solution

$$\chi = -\frac{\sqrt{2m}}{ik - m}$$

Inserting this back into (B.1) gives the required expression for a chiral Weyl fermion passing through a Majorana impurity. Taking the limit $\epsilon \to 0$, we find that $\psi$ jumps by a phase as it passes through the origin

$$\xi(x) = e^{ikx} \begin{cases} 1 & x < 0 \\ \frac{ik+m}{ik-m} & x > 0 \end{cases}$$

High energy modes, with $k \gg m$, are unaffected by the impurity. Low energy modes, with $k \ll m$, suffer a sign flip.

### The Spectrum on a Circle

To further understand the role played by the Majorana impurity, let us now consider a right-moving Majorana-Weyl fermion on a spatial circle, which we take to have length $L$.

We will impose periodic boundary conditions on this fermion, which means that it has a single Majorana zero mode. Such a system is anomalous and to rectify the situation we must add an odd number of extra Majorana modes. We do this by including $2n - 1$ Majorana impurities, at locations $x_i$ with couplings $m_i$. Periodicity of $\xi$ then imposes a quantisation condition on the momentum $k$ which is

$$\prod_{i=1}^{2n-1} 2 \tan^{-1} \left( \frac{m_i}{k} \right) = kL \mod 2\pi$$

When $m_i \ll 1/L$, the impurities pair up with the bulk zero mode to form $n$ independent complex zero modes. This results in a ground state degeneracy of $2^n$. Further modes are then quantised as $\sim 2\pi/L$.

Now consider increasing the interaction of a single impurity, say $m_1 \gg 1/L$. All bulk modes with $k < m_1$, including the bulk zero mode, undergo a sign flip, which means that their energy increases by $\pi/L$, corresponding to a spectral flow of $+1/2$. There are $n - 1$ remaining complex zero modes, and $2^{n-1}$ degenerate ground states.

Something a little different happens when we increase a second impurity coupling, say $m_2 \gg 1/L$. Once again, there is a spectral flow of $+1/2$. But instead of an impurity zero mode being lifted, it now mixes with a new bulk zero mode. Once again there are $2^{n-1}$ degenerate ground states. Clearly this pattern now repeats as further impurity couplings are increased.

### Absorbing Majorara Fermions into the Boundary State

The ideas described above help build intuition for how Majorana boundary modes can be incorporated in a boundary state. To illustrate this, consider a single Dirac fermion $\psi$ on an interval of length $L$. We impose vector boundary conditions at one end

$$\psi_L = \psi_R \quad \text{at } x = 0 \tag{B.2}$$

and axial boundary conditions at the other,

$$\psi_L = \psi_R^\dagger \quad \text{at } x = L \tag{B.3}$$

As explained in detail in [35], these two boundary conditions are mutually inconsistent in the sense that they result in a single Majorana zero mode in the bulk. Indeed, if we write

$$\psi = \xi_1 + i\xi_2$$

Then $\xi_1$ has a zero mode, while $\xi_2$ does not.

We now invoke the doubling trick, and view both fermions as chiral, living on a circle of length $2L$. The boundary conditions mean that $\xi_1$ is periodic, while $\xi_2$ is anti-periodic. To make the theory consistent, we add a single Majorana impurity, $\chi$, at $x = 0$. Now we have two options:

- We could couple $\chi$ to $\xi_1$. As we've seen above, the resulting spectral flow renders $\xi_1$ anti-periodic. The net effect is that the right-most boundary condition (B.2) is shifted from axial, to vector, but with a theta angle $\theta = \pi$, so that $\psi_L = -\psi_R$ at $x = L$. In this case, the ground state is non-degenerate. This shift of the theta angle due to a boundary fermion was also found in [3].

- If, instead, we couple $\chi$ to $\xi_2$, then the spectral flow renders $\xi_2$ periodic, with vanishing theta angle, so that $\psi_L = \psi_R$ at $x = L$. Now both $\xi_1$ and $\xi_2$ admit a Majorana zero mode, and there are two ground states.

## C    A D-Brane Perspective

The chiral boundary conditions have a more familiar interpretation in terms of boundary states for D-branes. Details of such states can be found, for example, in [49] or the textbook [50].

The geometric viewpoint arises after bosonization. This relates the $N$ Dirac fermions to $N$ periodic scalars, $\phi_i$ with the currents mapped as

$$\partial_+ \phi_i = \psi_i^\dagger \psi_i \quad , \quad \partial_- \phi_i = \bar{\psi}_i^\dagger \bar{\psi}_i$$

where $\partial_\pm = \frac{1}{2}(\partial_t \pm \partial_x)$. The chiral boundary conditions require that there is no net flow of the left- and right-moving currents $\mathcal{J}_\alpha$ and $\bar{\mathcal{J}}_\alpha$, defined in (2.4), into the boundary. In the bosonic picture, these become simple, linear boundary conditions on the periodic scalars

$$(Q_{\alpha i} + \bar{Q}_{\alpha i})\partial_x \phi_i = (Q_{\alpha i} - \bar{Q}_{\alpha i})\partial_t \phi_i \tag{C.1}$$

The trivial boundary condition $\mathcal{R} = \mathbb{1}$ gives Neumann boundary conditions $\partial_x \phi_i = 0$ in each direction, corresponding to a D-brane that wraps the full torus $\mathbf{T}^N$. Meanwhile, the other trivial boundary condition $\mathcal{R} = -\mathbb{1}$ gives a D0-brane, with $\phi_i = \text{constant}$. Clearly by taking $\mathcal{R} = \text{diag}(+1, \ldots, -1, \ldots)$ we have any D$p$-brane for $p = 0, \ldots, N$.

A general boundary state can be interpreted as a D-brane with flux, whose boundary conditions are written as

$$g_{\mu\nu} \partial_x \phi^\nu = B_{\mu\nu} \partial_t \phi^\nu$$

with $g$ the metric and $B$ the NS-NS 2-form.

The D-brane interpretation is particularly straightforward when $N = 2$ and we can consider the charge matrices (2.10) labelled by co-prime integers $p$ and $q$. The boundary conditions (C.1) are then

$$p\dot{\phi}_1 = q\dot{\phi}_2 \quad \text{and} \quad p\phi_2' = -q\dot{\phi}_1$$

This is simpler to interpret if we perform a T-duality on $\phi_2$, introducing $\partial_\mu \tilde{\phi}_2 = \epsilon_{\mu\nu} \partial^\nu \phi_2$. The boundary conditions then become

$$p\phi_1' = q\tilde{\phi}_2' \quad \text{and} \quad q\dot{\phi}_1 = -p\dot{\tilde{\phi}}_2$$

This describes a D-string wrapping $(p, q)$ times around the two cycles of the torus $\mathbf{T}^2$. Aspects of the boundary states for such a D-string, including the boundary central charge, were previously discussed in [51].

As described in Appendix A, the relevant boundary operators have dimension $L_0 = (m^2 + n^2)/(p^2 + q^2)$ for pairs of integers $m, n$. The associated RG flow describes the decay of a D-brane wrapping $(p, q)$ times around the torus to one wrapping $(m, n)$ times.

# Acknowledgments

We thank Nick Dorey for useful discussions. We're particularly grateful to Costas Bachas and Gerard Watts for useful comments on the manuscript. We are supported by the STFC consolidated grant ST/P000681/1. DT is a Wolfson Royal Society Research Merit Award holder and is supported by a Simons Investigator Award.

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
