# Peer review of "Boundary RG Flows for Fermions and the Mod 2 Anomaly"

_SciPost Physics Core_

## Round 2 · Referee Report · Anonymous (Referee 1) · 2020-10-23

Strengths

The introductory parts read like a well-prepared and well-presented lecture notes.

Report

This is a continuation of the authors' previous work 1912.01602, and studies the deformation by a relevant boundary operator of $U(1)^N$-preserving boundary conditions of $N$ Dirac fermions. The authors identified the endpoint of these flow within the class of $U(1)^N$-preserving boundary conditions, and identified the ratio of the boundary entropy with the square root of the dimension of the deforming operator. The (dis)appearance of Majorana zero modes at the boundary played an important role in their analysis.

The content is well-presented and the paper can be recommended for publication, once the optional changes suggested below are implemented.

Requested changes

  1. The authors identified the general relation $g_{IR}=r_{UV} \sqrt{\dim \mathcal{O}}$. How general do the authors think is this relation in general boundary conditions of general CFTs? Are there clear counterexamples where this relation does not hold?

  2. The authors say that $\mathcal{R}$ captures most of the properties of the boundary condition but a few extra information is necessary to fully specify it, for example the fermion vector $f$. Now, the authors determined $R_{IR}$ from $R_{UV}$ and the deforming operator $\rho$. What happens to the "extra information", for example $f$? Is there an equally nice formula specifying $Q_{IR}$ and $\bar{Q}_{IR}$ in terms of $Q_{UV}$, $\bar{Q}_{UV}$ and $\rho$?

  3. The referee did not get the joke on the famous hotel on p.7. Could the authors provide some more context?

  4. In Sec.3.3 the authors note the relation between $g^2$ and the number of "holomorphic superselection sectors". Is this relation special to this particular system, or is it applicable to more general 2d CFTs?

  5. on p.26, the authors discuss a flow from the Maldacena-Ludwig state to the Maldacena-Ludwig state, removing a boundary fermion. This sounded rather confusing to the referee, since the boundary state definitely changed. It would be less confusing to refer to this as a flow from a Maldacena-Ludwig state to another Maldacena-Ludwig state.

  6. on the same page, isn't a joke on Majorana presumably killed on a boat somewhat inappropriate for the living relatives of Majorana?

  7. In the Appendix C the authors discuss bosonized D-brane perspective. What happened to the fermion number operator $(-1)^F$, which played rather important roles in the discussion in the main paper? In the recent discussion of SPT phases and anomalies, the choice of the spacetime structure affects many things. Naively, we would like to treat fermions as spin theories and bosons as non-spin bosonic theories, and going back and forth between them in 2d requires care, as detailed e.g. in another paper by one of the authors (Tong) with Karch and Turner. How does that discussion combine with the analysis presented in this paper?

  8. on p.43, $p\phi_1=q\dot \phi_2$ in the second equation should be $p\phi_1'=q\dot\phi_2$.

---

## Round 2 · Referee Report · Anonymous (Referee 2) · 2020-11-19

Strengths

1 - The paper has a very striking new result linking the change in the g-value to the dimension of the perturbing operator which has not (I believe) been noticed before.

2 - The paper has extensive examples which help understand the problem and the result (not too many, a good number).

Weaknesses

1 - On a purely formal level, the paper does not have a conclusion section and so does not meet the criteria to be published as it is.

2 - The result relies heavily on the assumption that the $U(1)^N$ symmetry is restored at the IR fixed point. I did not see more discussion of this than the simple statement of the assumption. The results obtained with the assumption are consistent with the assumption, but that is a circular argument. I did expect non-trivial examples in which it is known to be the case. Are there any?

3 - There are a few places where assumptions or results particular to this model are not explained - this could confuse non-experts, either into believing that specific results presented here are general, or that there is no assumption being made. There are four in particular I thought should be changed. (a) It is important that the $U(1)$ charges are not degenerate - equivalently, the matrices $Q$ and $\bar Q$ introduced on page 9 should be invertible. This is implicit in the un-numbered equations after (2.2) but should be made explicit. This is related to (b) It is important that the ground states defined by (2.6) are unique. I think this should be stated. If not then (c) the form of the Ishibashi state in the un-numbered equation after (2.7) is only correct if the states satisfying (2.5) in each charge sector are unique, otherwise there is generically a mixing matrix between states of the same charge. Finally, just before section 2.2, I think readers should be reminded that it is quite possible to have boundary conditions with $g>1$ which are nevertheless stable - for example in the tri-critical Ising model. It is remarkable that in this case all such boundary conditions are unstable.

4 - In the un-numbered equation at the bottom of page 21, it is not the vacuum expectation value that is zero but the expectation value in the presence of the boundary. I think the notation is confusing here.

5 - In the general introduction, there are places where it is really unclear what is going on. For example, equation (1.1) states that a trace is a non-integer. I assume this is meant to demonstrate a paradox but it looks most odd. This should be re-written. Just before that it is stated through an argument that looks spurious that a single fermion would act on a space of dimension $\sqrt 2$. Since this patently cannot happen, the argument must be wrong. I would have liked to see a (proper) reference to the computation of the path integral for a single Majorana mode.

6 - I thought the comment on page 26 about Majorana and his presumed death at sea to be completely tasteless and out of place in a scientific paper.

7 - It would have ben really helpful if all equations were numbered, if only when trying to write this report.

Report

I think this paper adds quite a lot to the literature on boundary CFT and boundary flows. My comments are only about presentation and discussion. It is unfortunate that it does not meet the requirements of Scipost Core which state that to be acceptable, a paper must "6. Contain a clear conclusion summarizing the results (with objective statements on their reach and limitations) and offering perspectives for future work." With the changes requested below I am confident that I could recomment publications.

Requested changes

Please address (most of) the issues mentioned in the weaknesses:

1 - add a conclusion section

2 - discuss more the assumption of the symmetry restoration at the IR fixed point

3 - make a distinction between results which are generic to boundary CFT and results which are specific to the models here

4 - Please be clearer about the problems with the quantisation of a single fermion. The talk [6] is indeed good on this, but cannot the same results and methods be found in published papers.

5 - Please consider the style of the paper. It is very loosely written - I understand that David Tong has a great reputation as a lecturer and that his notes are admired, but sometimes I think the tone here is inappropriate (as in the aside about Majorana) in a paper.

6 - Please consider numbering all equations

---

## Editorial Decision

resubmitted